# MASIF: Meta-learned Algorithm Selection using Implicit Fidelity Information

**Tim Ruhkopf**                                      *t.ruhkopf@ai.uni-hannover.de*
*Institute of Artificial Intelligence*
*Leibniz University Hannover*

**Aditya Mohan**                                     *a.mohan@ai.uni-hannover.de*
*Institute of Artificial Intelligence*
*Leibniz University Hannover*

**Difan Deng**                                       *d.deng@ai.uni-hannover.de*
*Institute of Artificial Intelligence*
*Leibniz University Hannover*

**Alexander Tornede**                                *a.tornede@ai.uni-hannover.de*
*Institute of Artificial Intelligence*
*Leibniz University Hannover*

**Frank Hutter**                                     *fh@cs.uni-freiburg.de*
*Machine Learning Lab*
*Albert-Ludwigs University Freiburg*
*Bosch Center for Artificial Intelligence*

**Marius Lindauer**                                  *m.lindauer@ai.uni-hannover.de*
*Institute of Artificial Intelligence*
*Leibniz University Hannover*

**Reviewed on OpenReview:** *https://openreview.net/forum?id=5aYGXxByI6*

## Abstract

Selecting a well-performing algorithm for a given task or dataset can be time-consuming and tedious, but is crucial for the successful day-to-day business of developing new AI & ML applications. Algorithm Selection (AS) mitigates this through a meta-model leveraging meta-information about previous tasks. However, most of the available AS methods are error-prone because they characterize a task by either cheap-to-compute properties of the dataset or evaluations of cheap proxy algorithms, called landmarks. In this work, we extend the classical AS data setup to include multi-fidelity information and empirically demonstrate how meta-learning on algorithms' learning behaviour allows us to exploit cheap test-time evidence effectively and combat myopia significantly. We further postulate a budget-regret trade-off w.r.t. the selection process. Our new selector MASIF is able to jointly interpret online evidence on a task in form of varying-length learning curves without any parametric assumption by leveraging a transformer-based encoder. This opens up new possibilities for guided rapid prototyping in data science on cheaply observed partial learning curves.

## 1 Introduction

Data scientists typically spend most of their time with data engineering and thus often have less time for choosing a well-performing algorithm for a given task (Anaconda, 2020). This selection process is typically powered by both the experience of the data scientist w.r.t. the performance of algorithms on previous tasks as

well as a limited number of partial evaluations of the candidate algorithms on the task at hand. In doing so, a data scientist inherently carefully trades off (i) time invested into actively gathering performance information on the new task to make an informed decision and (ii) the regret in terms of performance incurred by possibly not selecting the best algorithm.

On one extreme end of this trade-off lie classical AS (Rice, 1976; Kerschke et al., 2019) approaches which characterize a task using handcrafted meta-features (Vanschoren, 2018) of the corresponding dataset. These features are supposedly informative w.r.t. the performance of an algorithm on that particular dataset and used to learn which candidate to recommend for a new task. In practice, such dataset meta-features may show limited association with the performance of the considered classes of algorithms (Pfahringer et al., 2000; J. Fürnkranz, 2001). In particular, so-called dataset meta-features, corresponding to cheaply-computable properties of the dataset (including simple, statistical, information-theoretic, complexity- and model-based properties (Vanschoren, 2018)), are known to be only indicative of algorithm performance to a limited degree (Pfahringer et al., 2000). Similarly, the correlation of landmarking meta-features, i.e., performances of cheap proxy algorithms computed on the new dataset, can be limited, as they hinge on how well the proxy algorithms' performances are associated with those of the pool of candidates. For instance, the performance of a decision stump might not be informative on how well a deep neural network will perform.

On the other extreme end of this trade-off lie existing learning curve approaches (Mohr & van Rijn, 2022), which exclusively invest time into gathering premature performance approximations on the candidates and seek to extrapolate off of it. Examples of these approximations are validation performances of models trained for a limited amount of epochs or only on parts of the training data. As such, learning curve approaches invest comparatively more budget on obtaining information naturally associated with the final performance of the classes of algorithms. A common downside to these approaches lies in their extrapolation strategy. Observing only premature approximations of the performance data with little or no meta-knowledge about an algorithm's learning behavior, learning curve methods often resort to strong parametric assumptions (Mohr & van Rijn, 2022) and are myopic in the sense that an extrapolated estimate of the final performance can only be based on the observed premature performance values. Further, with few exceptions, such as (van Rijn et al., 2015; Klein et al., 2017b; Baker et al., 2018; Long et al., 2020), they only extrapolate each algorithm independently, ignorant of possible existing relations in learning behaviors to other algorithms.

In this work, we propose MASIF (Meta-learned Algorithm Selection using Implicit Fidelity)[1], an approach designed to support data scientists in the AS process in a much more intuitive manner. It supports a data scientist's active discovery process on a new dataset, as it allows refining the expectation on the final ranking over algorithms based on their incremental budget allocation. This provides the user with full flexibility in different trade-offs between invested budgets and regret of the final ranking. It is however also a limitation in the sense that a user has to make this decision. As alleviation, we later provide an exemplary solution to this but note that it opens up a new area for research.

MASIF is powered by a transformer-based model accounting for the sequential nature of learning curves and interpreting their (partial) informational content. It learns to interpret them based on the meta-knowledge w.r.t. the candidates' learning behavior obtained from observing them on other datasets. Using this encoder and the available meta-knowledge, our model neither requires explicitly extrapolating the curves nor does it necessitate assumptions on their parametric shapes. Learning a latent representation of each curve respectively, a subsequent transformer and MLP merge the gathered test-time evidence in the form of observed premature performances. This allows learning cross-correlations between the algorithms' learning behaviors (cf. Figure 1).

Overall, we make the following contributions:

1. From a practical perspective, we formalize a meta-learning multi-fidelity setup that closely resembles a data scientist's workflow. Based on this we discuss the budget-regret trade-off that contemporary algorithm selectors implicitly make.

2. We introduce MASIF, a novel learning curve interpreter living in this framework, that leverages both dataset meta-features and formulates a well-informed meta-prior on the candidates' learning

---

[1]MASIF's code is published on `https://anonymous.4open.science/status/MASIF-824D`

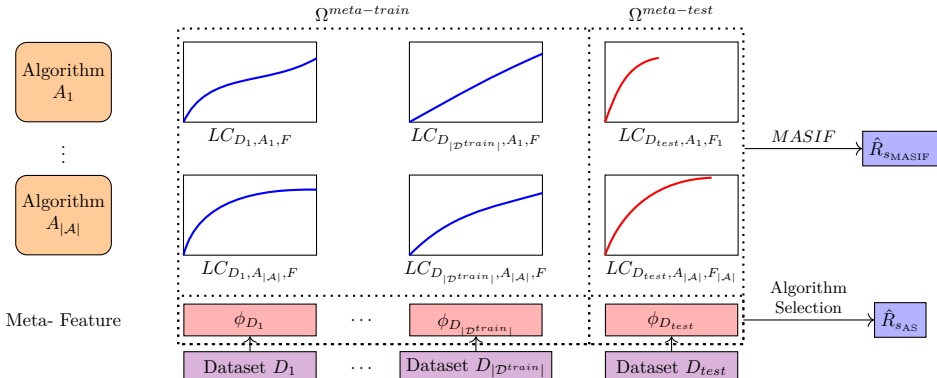

Figure 1: In contrast to classical algorithm selection trained only on meta-features, MASIF utilizes the information from both meta-features and learning curve values to predict the best performing algorithms.

behaviors. It mitigates the fallacies of both, classical AS and learning curve-based approaches by interpreting multiple learning curves of varying lengths and, utilizing meta-knowledge to combat myopia[2], leading to a better-informed ranking with less compute on a new dataset.

3. We propose a new evaluation scheme based on fidelity slices suited for demonstrating the aforementioned trade-off.

4. We demonstrate the usage of our novel learning curve interpreter as an algorithm selector based on a user's budgetary and regret preferences, and evaluate its performance against other baselines on several benchmarks.

## 2 Preliminaries

As we seek to unify classical AS, multi-fidelity, and learning curves, we re-examine them and offer a notation, with which we later detail how selectors behave. Particularly, this notation highlights both the available meta-knowledge and the associated cost of acquiring information on a new dataset.

**The Algorithm Selection Problem** denotes the task of finding an algorithm selector $s : \mathcal{D} \to \mathcal{A}$ from the space of algorithm selectors $\mathcal{S}$, which selects the presumably most suitable (machine learning) algorithm $A^* \in \mathcal{A}$ for a given dataset $D \in \mathcal{D}$ where $\mathcal{D}$ is a set of datasets, also called meta-dataset. Here, suitability is expressed through a costly-to-evaluate performance metric $m : \mathcal{D} \times \mathcal{A} \to \mathbb{R}$, such as accuracy or cross-entropy on the test data, quantifying how well the given algorithm performs on the given dataset *at the end* of its training process. The (hypothetical) best algorithm selector, also called the oracle, $s^*$, chooses the best algorithm for every dataset $D$ on the final performances:

$$s^*(D) = A^* \in \arg\max_{A \in \mathcal{A}} m(D, A) \tag{1}$$

**Classical Algorithm Selectors** seek to meta-learn which algorithm will perform best on a new dataset. Since the performance of an algorithm $A \in \mathcal{A}$ is costly to evaluate, a complete enumeration over $\mathcal{A}$ to select the best one is often not feasible in practice, which leads to the application of machine learning to predict the best algorithm. To allow learning such a selector based on meta-training data, datasets $D$ are characterized by their dataset meta-features $\phi_D$ and are fed as input to the selector. In the classical setting, these are quickly-computable properties of a dataset, called dataset meta-features Vanschoren (2018) such as, e.g., the number of data points, number of explanatory features or the entropy of a dataset's dependent variable.

---

[2]We use the term myopia – synonym for short-sightedness to highlight the limited information horizon. In particular, these kinds of predictors/selectors can base their extrapolation only on the observed part of a curve and are ignorant w.r.t. meta-knowledge.

For the purpose of analyzing a selector's associated cost, computing $\phi_D$ is part of the selector's computational budget on a meta-test dataset to gain information about it. Optionally, $s$ can also take algorithm meta-features $\phi_A$ as input, e.g., the number of neurons in a particular layer of a neural net (Tornede et al., 2020; Pulatov et al., 2022).

In practice, many AS approaches, either implicitly (Xu et al., 2012; Amadini et al., 2014) or explicitly (Cunha et al., 2018; Hanselle et al., 2020), compute a ranking over the algorithms $\mathcal{A}$ and return the best one. Correspondingly, as we also rely on rankings – for the remainder of this paper – we assume that a selector returns a ranking from the ranking space $\mathcal{R}(\mathcal{A})$ over the discrete set $\mathcal{A}$, i.e., we consider algorithm selectors of the form $s : \mathcal{D} \to \mathcal{R}(\mathcal{A})$.

In the classical setting, the data used to train a selector is called meta-training set[3] to differentiate it from the actual datasets $D$, on which the individual algorithms are trained and evaluated. It is defined as

$$\Omega^{meta-train} = \{\phi_D, \phi_A, m(D, A) | D \in \mathcal{D}^{train}, A \in \mathcal{A}\}, \tag{2}$$

where $\mathcal{D}^{train}$ is a set of datasets to meta-train on. $\mathcal{D}^{train}$ is used to train a selector $s$ to predict rankings over the known set of algorithms $\mathcal{A}$ across datasets with minimal meta-train loss $\mathcal{L} : \mathcal{R}(\mathcal{A}) \times \mathcal{R}(\mathcal{A}) \to \mathbb{R}$. It measures the error between the predicted ranking $s(D)$ and the ground-truth ranking $R_D^*(\mathcal{A}) \in \mathcal{R}(\mathcal{A})$. Hence, we solve the optimization problem defined as

$$\min_{s \in \mathcal{S}} \sum_{D \in \mathcal{D}^{train}} \mathcal{L}\big(R_D^*(\mathcal{A}), s(D)\big). \tag{3}$$

Similarly, the meta-test loss $\mathcal{L}'$ of $s$ is measured on the known set $\mathcal{A}$ on hold-out datasets $D \in \mathcal{D}^{test}$.

$$\mathcal{L}^{test}(s) = \sum_{D \in \mathcal{D}^{test}} \mathcal{L}'(R_D^*(\mathcal{A}), s(D)). \tag{4}$$

While $\mathcal{L}$ usually is a differentiable training loss, during test time $\mathcal{L}'$ does not need to have this restriction. For the remainder, we assume this to be a regret function. In particular, since algorithm selectors return a recommendation of the most promising algorithms on the given dataset, we use the top-$k$ regret for the test-loss as defined in Eq. (4)

$$\mathcal{L}_k^{test}(R_D^*(\mathcal{A}), s(D)) = \min_{A \in \text{top-}k(s(D))} m(D, A^*(D)) - m(D, A), \tag{5}$$

where the top-$k$ operator returns the first $k$ elements of the ranking that $s$ recommended and $A^*(D) = \arg\max_{A \in \mathcal{A}} m(D, A)$. The selector $s$ is allowed to access the available test-time information $\{\phi_D, \phi_A | A \in \mathcal{A}\}$ for its prediction on $D$, resulting in the meta-test knowledge: $\Omega^{meta-test} | D = \{\phi_D, \phi_A | A \in \mathcal{A}\}$.

A particularly strong kind of meta-feature in classical AS is landmarking (Pfahringer et al., 2000). Landmarking features report the validation performances $m(D, A)$ of cheap algorithms that are usually not in the pool of candidates ($A \notin \mathcal{A}$) as dataset meta-features. This relaxes the assumption that meta-features need to be very fast to compute.

The utility of classical and landmarking meta-features w.r.t. the selection task depends on the alignment of their description of the topology to the topology that the candidate algorithms' inductive biases are successful on – as observed in terms of their final test performance. This is particularly true for landmarking features as they apply their inductive biases to the datasets' topology. Their relevance is related to the overlap in the functions that both sets of algorithms describe. For example, a shallow decision tree might be partially informative regarding the performance of other tree-based algorithms, such as random forests (Breimann, 2001) or xgboost (Chen & Guestrin, 2016). We refer to the question of information content as the *alignment problem* of meta-features and algorithm performance.

---

[3]We refer to this meta-training set as meta-knowledge interchangeably.

**Multi-Fidelity Optimization**    Since training a machine learning model to completion can be very expensive, multi-fidelity optimization (Li et al., 2018) aims at using cheap-to-evaluate proxies to make efficient decisions, e.g., which algorithm performs best. In automated machine learning (AutoML, Hutter et al. (2019)), these proxies can include training for a limited number of epochs or training on a subset of the training data.

Formally, we seek to approximate a costly-to-evaluate function $F$ by a cheaper fidelity $f_i \in \mathcal{F}$ from the space of fidelities $\mathcal{F}$. We note that also $F$ is part of $\mathcal{F}$. Each fidelity $f_i$ comes with a cost $c(x, f_i)$ to evaluate $f_i(x)$ at point $x$. A common assumption is that the approximation quality of $f_i$ w.r.t. $F$ gets better in proportion to its cost $c(\cdot, f_i)$, leading to a trade-off between approximation quality and cost of the fidelity.

**Learning Curves**    form an ordered sub-case in multi-fidelity optimization, that is $\mathcal{F}$ is cost-ordered. In our case, we consider a learning curve of an algorithm to be the performance metric $m_{lc} : \mathcal{D} \times \mathcal{A} \times \mathcal{F} \rightarrow \mathbb{R}$ observed on cost-ordered fidelities $f_1, \ldots, F_A$, where $c(x, F_A) \leq c(x, F)$ and $F_A$ is the maximally observed fidelity for algorithm $A$. Notice, that the definition of $m_{lc}$ generalizes the former notion of $m$ to be observable at multiple fidelities. Formally, a learning curve can be defined as

$$LC_{D,A,F_A} = (m_{lc}(D, A, f))_{f=f_1,\ldots,F_A} \ . \tag{6}$$

Notably, we distinguish between a partial learning curve, i.e., $F_A \neq F$, and a full learning curve, i.e., $F_A = F$. Each algorithm may be observed at a different maximum fidelity $F_A$, since a user may decide to spend different amounts of computing resources to evaluate different algorithms $A$.

## 3    MASIF: Interpreting Partial Learning Curves Jointly

In this section, we introduce our core contributions: (i) extending the classical AS setup by adding multi-fidelity information, allowing for a more detailed analysis of the budget-regret trade-off of algorithm selectors; (ii) our transformer-based MASIF approach as an efficient method for this extended data setup; and (iii) a data augmentation scheme to achieve strong predictive performance and enable MASIF to accept arbitrary budget allocation strategies.

### 3.1    Extending Algorithm Selection by Multi-Fidelity Information

In addition to dataset meta-features, we propose to use information from learning curves as multi-fidelity information for AS. This provides a new source of information by prematurely inquiring about a new dataset's topology from the perspective of the candidates' inductive biases that is both relatively cheap and surely well-aligned. We summarize this setup in Figure 1.

**Meta-Training Data**    The aforementioned idea extends our available training *meta-knowledge* in Eq. (2) and Eq. (3) by fidelity information, amounting to

$$\Omega^{meta-train} = \{\phi_D, \phi_A, LC_{D,A,F} | D \in \mathcal{D}^{train}, A \in \mathcal{A}\} \tag{7}$$

where the learning curves $LC_{D,A,F} \in \mathbb{R}^F$ replace the performance values $m(D, A) \in \mathbb{R}$ in Eq. (2). For our experiments, we observed full learning curves up to the maximal fidelity $F$ for the meta-training dataset. The meta-training dataset encodes several aspects of the available meta-knowledge: (i) the relation of the algorithm's performance w.r.t. the dataset's topology, (ii) how information about the algorithm's inductive bias is unrolled across fidelities marginally, and (iii) how the learning behavior of different algorithms relate to each other on the datasets they are applied to.

**Meta-Testing Data**    The meta-testing phase provides *partial* learning curves observed on a new hold-out dataset. The set of curves provides evidence of the dataset's topology observed through the lens of the algorithms. The available information at test time for our selector $s$ in Eq. (4) is:

$$\Omega^{meta-test} = \{\phi_D, \phi_A, LC_{D,A,F_A} |, F_A \le F, A \in \mathcal{A}, D \in \mathcal{D}\}. \tag{8}$$

Notably, depending on a user's budgetary preferences, some of the learning curves may be revealed nearly entirely or not at all; in general, we assume that the set of all learning curves may be incomplete. Since an algorithm selector returns a ranking over the algorithms on the given dataset, we use the top-$k$ regret for the test-loss on the final fidelity $F$ as defined in Eq. (4).

## 3.2  MASIF Architecture

We choose a deep learning architecture because of its functional flexibility and its non-parametric modelling. However, we have to exploit the little data typically available in AS without grossly over-fitting to the meta-data. Our proposed architecture is displayed in Figure 2 and consists of five main building blocks:

**Block 1: Dataset meta-feature embedding**  Assuming some valuable information on the topology in the classical meta-features $f_D$, an MLP encodes them into a latent space. Utilizing this cheap set of features, from an informational standpoint, our method can only improve over classical AS methods and further allows us to contextualize the observed learning curves to the observed dataset. If, for instance, $\phi_D$ conveys some information regarding the complexity of the dataset, this may be indicative of how fast an algorithm might learn, affecting the shape of the curve.

**Block 2: Learning curve preprocessing**  As preprocessing of the learning curves, we apply zero-padding to fill the learning curves up to the final fidelity $F$ and accordingly generate a mask to store the padding information. To avoid technical issues with fully masked sequences, we append an extra learnable End Of Sequence (EOS) token (Dosovitskiy et al., 2020). We then apply a shared MLP separately to each position of each learning curve. Embedding each position of a learning curve individually increases the expressivity of our one-dimensional learning curve sequences into $e$ dimensions akin to BERT's vector representations for each word (Devlin et al., 2019). To overcome the transformer's native positional ignorance, we follow Vaswani et al. (2017) in encoding the positional information with sine and cosine functions and adding these values to the embedded learning curves. This is a crucial step in preserving the sequence information, which in our case conveys the order in learning curves.

**Block 3: Learning curve embedding**  We use a variation of the transformer-encoder proposed by Vaswani et al. (2017) on the set of partially observed learning curves $LC_{D,A,F_A}$ of each algorithm $A \in \mathcal{A}$. This translates the (incomplete/masked) learning curve sequence into a latent vector representation enriched with the meta-knowledge of the algorithms' marginal learning behaviors.

Since the classical dataset meta-features may carry valuable information (if they are available) for the amount and variability of information at each position in the sequence, we introduce a mechanism we dub *guided attention*. Similar to the classical attention module, our encoder receives a query $Q$, key $K$, and value $V$ tensor, containing the preprocessed learning curves from Block 2. We multiply the query $Q$ of the attention heads element-wise with a linear projection of the embedding described in Block 1 if dataset meta-features are available. The projection's role is merely to fit the shape of $Q$. To increase parallelism and reduce the number of learnable parameters, we use a batch trick by only processing one dataset at a time, but conceive the set of algorithms $\mathcal{A}$ to be a batch of dimensions $[\mathcal{A}, \mathcal{F}, e]$, that is processed in parallel. The attention-guided transformer-encoder layer is repeated $N$ times to yield a marginal summary of the latent representation of each of the learning curves. The order of the batch is held fixed over all datasets, s.t. the subsequent module receives the latent vector representation of an algorithm's learning curve at the exact same position every time.

Given the generated feature maps, we reduce the $[\mathcal{A}, \mathcal{F}, e]$-dimensional tensor along the fidelity dimension, using a learnable weighted average over the fidelities $\mathcal{F}$ applied independently over $\mathcal{A}$, resulting in $e$-dimensional vectors. This formulates a joint representation of the evidence on $D$ encoded in these vectors. Given the properties of the target dataset, the model should focus on different stages of the learning curves. Therefore, we make these weights learnable. More precisely, if dataset meta-features are available, these weights are

computed by an MLP that projects the dataset meta-feature embedding to the size of $\mathcal{F}$. Otherwise, these weights directly become learnable parameters.

**Block 4: Fusing the evidence.** The model up until this point is still unaware of the cross-algorithm information between different learning curves, which is essential to predict the relative ranking of each algorithm. Therefore, we build another transformer on top of the first transformer that fuses the evidence on the topology derived from the partial learning curves of all $A \in \mathcal{A}$. The rationale is that since the algorithms are always presented in the same position, the position of these vectors holds meaning. From the perspective of BERT, each algorithm vector is the embedding to a token, and the sequence of algorithms is a sentence. Therefore, we transpose the set of representations and make the batch dimension explicit $[1, A, e]$. To let the transformer identify different algorithms, we encode the algorithms' features (i.e., their hyperparameters or a simple one-hot encoding when no hyperparameter is available) as positional encoding and attach them to the generated transformations. The rationale for treating the algorithm meta-feature embedding as a positional encoding – if corresponding features are available – is to give the learning curve embeddings more context at best and provide an alternative index that replaces the one-hot encoding at worst. Treating these algorithm representations as positional encodings should aid the second transformer encoder in how to fuse the sources of information. We have added an ablation on this design choice in Appendix 8.

**Block 5: Ranking** Yet again, we employ an MLP on the concatenation of the vectorized evidence and the dataset meta-feature embedding to fuse their information and project to the size $n_{\mathcal{A}} = |\mathcal{A}|$. The returned values are scores for the respective algorithms, corresponding to latent utility values, which are used to construct a ranking across the algorithms. As an objective, we would like to maximize the Spearman rank correlation between predicted and true algorithm ranks. In order to obtain a differentiable loss, we thus leverage a differentiable sorting algorithm (Blondel et al., 2020) to compute the training loss.

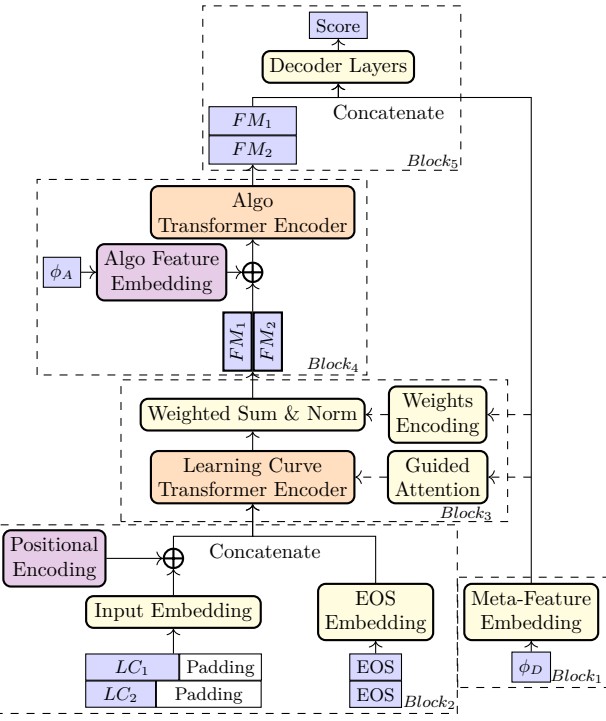

Figure 2: MASIF Architecture. Blue rectangles represent data, yellow rectangles denote MLP layers, purple rectangles indicate positional encodings, and orange rectangles are transformer layers. MASIF receives the partial learning curves of all the algorithms $LC_{D,A,F_A} \forall A \in \mathcal{A}$ (in this figure, we only show two learning curves, i.e., $LC_1$ and $LC_2$, indicating $LC_{D,A_1,F_{A_1}}$ and $LC_{D,A_2,F_{A_2}}$). The two transformer encoder layers apply attention operations on the fidelity and the algorithm dimensions, respectively. Therefore, the output of each transformer needs to be transposed to fit the dimension. The output of the last transformer ($FM_1$ and $FM_2$) is concatenated with the embedded meta-features $\phi_D$. Finally, the joint features are fed to the decoder to get the predicted ranks.

### 3.3 Data Augmentation

Since AS usually only has a limited amount of dataset-algorithm combinations in its meta-training dataset for which we observe a single learning curve instantiation, respectively, we use a data augmentation strategy. We mask the learning curves by sampling the available learning curve length $F_A$ uniformly and independently from $[0, F]$, where 0 indicates a complete lack of information on that learning curve. In addition to circumventing meta-overfitting, this strategy allows our model to be presented with any budget allocation (combination of fidelities), i.e. $\{F_A \in \mathcal{F} \cup \{0\}\}_{A \in \mathcal{A}}$ – including the option of not evaluating a curve during test-time.

### 3.4 Hyperparameters

To foster reproducibility, we list the following hyperparameter choices in the above architecture. Details regarding their choices, including ablations, can be found in Appendix A.4. We encode all the meta-features (i.e., $\phi_D$ and $\phi_A$) with a 2-layer MLP (with hidden size of 128 and 64) into an embedding of size 64. All transformer encoder layers in MASIF have the same architecture with hidden size 128 and 4 attention heads. Each of the transformer encoders (Learning Curve Transformer Encoder and Algo Transformer Encoder) have 2 transformer layers are applied with a dropout rate of 0.2. We train MASIF with an ADAM optimizer with a learning rate of 0.001 and beta values as 0.9 and 0.999 for 500 epochs while neither learning rate scheduler nor weight decay are employed.

### 3.5 MASIF's Benefits

The transformer encoding of the learning curves offers significant advantages: at no point do the learning curves require extrapolation. Instead, only their informational content is summarized and transformed into a conditioned ranking score. Our method, therefore, does not fall victim to error propagation as often encountered in time series regression or multi-label classification (Senge et al., 2012). Additionally, leveraging the meta-knowledge obtained by observing the learning curves of the same algorithms on meta-training datasets, we combat multi-fidelity's inherent myopia problem effectively and can be fully non-parametric in the shape of the learning curves due to a strong meta-prior encoded in the weights of our architecture. To this end, MASIF exploits the available meta-knowledge extensively of having observed the algorithms' learning curves on multiple datasets jointly. Depending on the type of algorithms and fidelity, this meta-knowledge in the form of algorithms' learning curves comes at almost no cost when collecting the meta-training data in the classical AS setting. Crucially, taking a multi-fidelity lens – independent of whether or not we include classical and landmarking meta-features – its ranking ability is no longer bounded by the alignment of tediously handcrafted and highly task-specific meta-features. Instead, it is now governed by the trade-off between the approximation quality of the fidelities and the induced cost of querying them. It does, however, readily consider expert knowledge in form of dataset meta-features to contextualize the observed learning curves.

Given some budget allocation, the MASIF model corresponds to an algorithm selector since it produces a ranking that takes all the available meta-knowledge and the gathered evidence on the test dataset into account. However, since any budget allocation is possible, it can be readily utilized not only as a selector but as an interpreter of the incrementally arriving performance feedback from the budget allocation. Queried continuously as more evidence arrives, users can observe the shift in expectation on the final ranking, conditioned on what has been observed so far. This application mode makes this model an actionable support system for data science practitioners facing the AS problem.

The trade-off between approximation quality and budget costs opens up the opportunity of defining schedulers that successively queries the fidelities of algorithms (i.e., extending their learning curves) by exploiting our model. Since our contribution focuses on the model part and since MASIF's model works with arbitrary strategies for deciding the budget allocation, in our experiments we show exemplary results with the prominent scheduling approach of *Successive Halving (SH)* (Jamieson & Talwalkar, 2016). While the original SH procedure in each "bracket" retains the current top-performing algorithms, our variant, dubbed SH Scheduler retains those that based on the current observation MASIF predicts to be top-performing on the final ranking.

## 4 Experiments

In this section, we first introduce a new evaluation protocol for our proposed extension of the classical AS meta-data setup with added fidelity information. Then, we detail our baselines along with their perks and shortcomings, briefly describe the meta-datasets we use as benchmarks and subsequently detail our findings.

### 4.1 Fidelity-Slice Evaluation Protocol

In order to assess the ability of methods to improve performance as a function of the length of the observed learning curves, we introduce a standardized *fidelity-slice evaluation protocol* that demonstrates how a method's top-$k$ regret w.r.t. the true ranking changes as more and more fidelity information becomes available. This way, we can observe the progression of an information-bounded expectation traded with the additional cost incurred. In particular, this protocol computes the regret of $s(D)$'s prediction on the test set at a gradually increasing amount of available fidelity slices; i.e. all algorithms are observed up to the same fidelity.

### 4.2 Baselines

We consider several baselines to highlight different aspects of our setup and architecture ranging from simple random rankings over classical AS to an extended version of successive halving.

**Random Baseline.** Consider a selector that randomly guesses the ranking. It will lead to a regret distribution depending on (i) $k$ in top-$k$ (ii) the spread and clustering of learning curves on a single dataset, indicating the hardness of the ranking task and (iii) the performance scale. This baseline averages the obtainable regret of each dataset instance. It contextualizes the selectors' regrets across tasks.

**Classical Algorithm Selection.** While AS allows for generalization over dataset instances, its limitation in ranking performance lies in the limited expressivity and relevance of the employed meta-features. Therefore, we seek to demonstrate the benefit of partial learning curves as an addition to classical meta-features. For that purpose, we compare against SATzilla'11 (Xu et al., 2012), a portfolio-based algorithm selector, which models the AS problem as a multi-class classification problem and solves it using a cost-sensitive all-pairs decomposition to single label classification by employing one random forest classifier for each subproblem. The final ranking is then obtained by voting. This selector is irrespective of fidelity and will produce a constant value in the slice evaluation protocol.

**Fidelity enabled Algorithm Selection.** Classical AS only considers the dataset and algorithms' meta-features. Considering that our method's learning curve embedding could be conceived to be a latent meta-feature $\phi_D$, classical AS can conceptually similarly be enabled. To achieve that, the classical $\phi_D$ can be extended by $m(D, A, f)$ for some intermediate fidelities $f < F$ observed for all $A \in \mathcal{A}$. Given $f \in \mathcal{F}'$ with $\mathcal{F}' \subset \mathcal{F}$, this implies that $|\mathcal{A}| \cdot |\mathcal{F}'|$ columns are added to the feature space. Consequently, we limit $|\mathcal{F}'|$ to a subset of evenly spaced fidelities.[4] In the slice evaluation protocol, the intermediate fidelities are added only up until the horizon of the protocol.

**Parametric Learning Curve Predictors.** Neglecting any meta-knowledge and combating myopia through strong parametric assumptions, a naive learning curve predictor can interpret any partial information available. Such a predictor fits a parametric curve to each partial curve and extrapolates it to the final performance. The selector $s$ independently extrapolates to the final performances and ranks the algorithms according to these. To create a meaningful and more expressive selector that mimics a practitioner's attempt at extrapolating learning curves for AS, we fit all parametric curves described in Mohr et al. (2022) and select the best fitting one on the observed part for extrapolation. This baseline highlights that learning curve predictors are used as algorithm selectors by practitioners when lacking meta-knowledge. Surpassing this baseline indicates that our method's expectation of the ranking can benefit from joined meta-knowledge about an algorithm's past progressions and related learning behaviors.

**LCNet.** Learning curve prediction with Bayesian neural networks Klein et al. (2017b), originally designed for hyperparameter optimization problems can be applied to our algorithm selection task whenever hyperparameters or algorithm meta-features are available on a benchmark. The underlying Bayesian Neural Network takes the hyperparameter configuration and the fidelity level at which it is observed as input and outputs both a variance and a mean prediction. The mean is built of a weighted ensemble of the parametrizations for a fixed set of parametric learning curves. It is a meta-agnostic but expressive parametric baseline.

---

[4]In our experiment, this amounts to the fidelity sequence $[0, 0.2, 0.4, 0.6, 0.8, 1]$.

**IMFAS.** Implicit Multi-Fidelity Algorithm Selection (IMFAS) (Mohan et al., 2022) utilizes an LSTM-based architecture that initializes its hidden state using an MLP encoding of the dataset meta-features and auto-regressively accepts slices of fidelity to refine the expectation of the ranking.

**Successive Halving (SH).** Despite its simplicity, SH (Karnin et al., 2013; Jamieson & Talwalkar, 2016) has several desirable properties to compare against in terms of its interpretation of partial learning curves. It is a meta-agnostic and therefore myopic, fidelity-aware but non-parametric method. SH naturally produces a myopic ranking, by the level at which it terminates the algorithm. Ties within this set of terminated algorithms are resolved by their relative performances on that bracket. Limiting the fidelity information horizon of the selectors implies that for every such horizon, we need to recompute the ranking induced by SH. Notably, to permit SH the same information horizon, the last available fidelity in its schedule will be the maximum fidelity in that horizon. The benefit of this convention is that SH has access to the same fidelity information and can make better decisions for the algorithms it recommends. Its drawback of SH as a baseline is that the incurred cost differs from that of our method.

### 4.3 Datasets

Where possible, we utilize training, validation, and test learning curves for the training of all approaches. These are obtained by splitting each dataset in the corresponding benchmark in training, validation, and test data, training the corresponding algorithms on the training data under each fidelity, and computing their validation and test performance corresponding to the model trained for that fidelity. While the classical AS approaches only receive the final test performance on the meta-train datasets as training data, we provide (partial) validation curves to SH, the learning curve approaches, and MASIF to train on, but measure the final regret with respect to the full fidelity test performance. The same applies when the approaches are applied to new test datasets. We then assess the performance of each approach by performing a 10-fold outer cross-validation of the meta-dataset. We use three different benchmarks (more details in Appendix A.1):

- **Synthetic.** We constructed the Synthetic meta-dataset based on parametric learning curves taken from Mohr et al. (2022), specifically to demonstrate the myopia of SH and learning curve predictors. Particularly, it introduces noisy curves, that exhibit crossing points. We show how MASIF alleviates myopia through strong prior knowledge w.r.t. the functional family an algorithm is adhering to.
- **Task-set.** Task-set (Metz et al., 2020) is a fairly noisy real-world dataset based on parameterizations of the Adam optimizer (Kingma & Ba, 2015), with a variety of learning rates, run on a large variety of modern deep learning architectures and datasets.
- **Scikit-CC18.** With Scikit-CC18, we evaluated multiple well-known scikit-learn (Pedregosa et al., 2011) algorithms on the classification benchmark OpenML-CC18 (Bischl et al., 2021b).
- **LCBench.** LCBench (Zimmer et al., 2021) is a learning curve benchmark of different funnel-shaped neural networks and hyperparameters for tabular data. We sub-sampled the set of configurations to only include 170 combinations as algorithms to be selected.

These four benchmarks differ in their availability of dataset and algorithm meta-features: while Synthetic and Task-set exhibit neither, Scikit-CC18 and LCBench exhibits both. Since Scikit-CC18 and LCBench are the only benchmarks with available dataset meta-features $\phi_D$, we can only evaluate SATzilla and IMFAS on them. Similarly, the dataset and algorithm meta-features can only be exploited by our architecture on these benchmarks.

### 4.4 Results

Summarizing our results, depicted in Figure 3, on benchmarks Task-set and Synthetic, MASIF outperforms the other methods in terms of top-1 regret, and on Scikit-CC18 it performs competitively. Additional results on an NLP subset of Task-set can be found in Appendix A.2, as they show the overall same tendencies. Since our baselines applied in the slice-evaluation protocol on their own already reveal a few features of the meta-datasets we would like to detail their implications first. Afterwards, we discuss our experiments in light of the available meta-knowledge w.r.t. myopia and parametric assumptions.

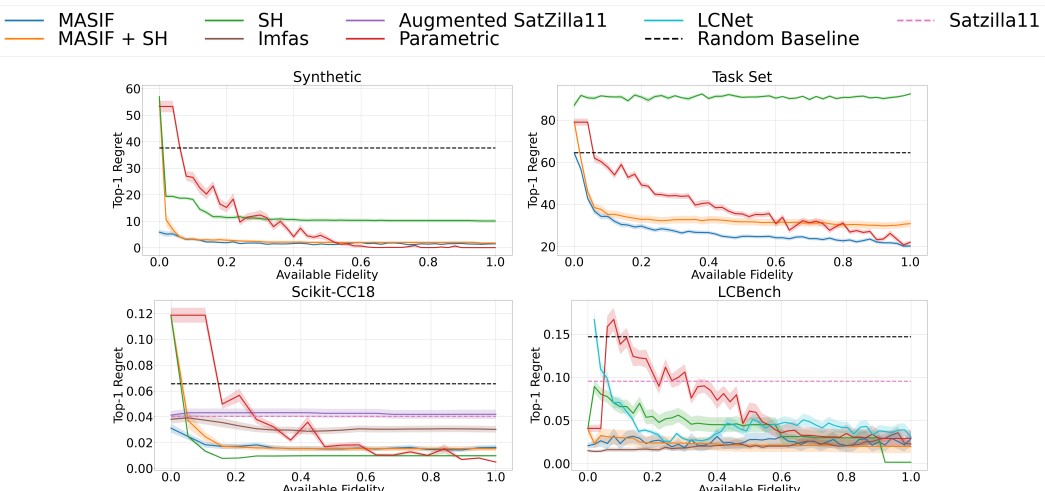

Figure 3: Average test-set top-1 regret over ten meta-dataset folds in the slice evaluation protocol. Available fidelity is expressed in the share of the target fidelity's budget. The standard error bands originate from five repetitions for each of the ten folds. Note: Scikit-CC18 and LCBench exhibit dataset meta-features, while Synthetic and Task-set do not. Task-set is reported for the image data subset.

**Benchmark Difficulty.** First of all, paying close attention to the random baseline and SH already indicates the difficulty of a benchmark; the random baseline can be thought of as an upper regret bound for any meaningful selector. If this threshold already incurs a small regret as in Scikit-CC18, this implies that there is only little to gain, increasing the difficulty for a selector to improve. SH, on the other hand, pays close attention to the earlier parts of the curve. If this is already very indicative in terms of the final performance, only few crossings of the curves occur and thus decisions based on low-fidelity are good w.r.t. final performance. In this sense, Scikit-CC18 is an arguably easier meta-dataset than Task-set, as SH is close to oracle performance after a few fidelity slices in their top-3 regrets (compare with Figure A.3). SH's detrimental performance on Task-set on the other hand likely is due to the high variability and crossing of the curves, indicating the difficulty of this meta-dataset. The fact that no method obtains zero regret on Task-set at the final fidelity originates from the noisy target as indicated by the final fidelity of the parametric baseline. Moreover, parametric learning curve is (almost) guaranteed to yield oracle performance on Synthetic by construction. Inspired by Multi-LCNet (Jawed et al., 2021), which uses LCBench as its sole benchmark, we ran our experiments on LCBench as well. However, we find that LCBench is not a strong benchmark for our setup, due to (i) the confidence bounds overlap (ii) the very small regret of around 0.02 of most approaches and (iii) the fact that all of the meta-aware selectors do not require additional fidelity information to adjust their prior. This is additionally supported by (i) SH's strong low-fidelity performance with only little gain by fidelity and (ii) both SH's and the parametric learning curve baseline's strong final performances.

**The Role of Meta-knowledge.** We highlight the fact that zero-fidelity corresponds to the classical AS setup and expresses the prior belief of a selector in the final performance based on the meta-knowledge. Baring this in mind, the fact that MASIF is almost constant and already top-performing on Synthetic indicates that the strong prior derived from meta-knowledge is already sufficient – which in the case of Synthetic is by construction. A decrease in regret indicates that by adding additional test-time fidelity information up to $F_{max}$ refines this meta-knowledge-based belief.

The trajectory of the parametric learning curves is a strong myopic baseline to demonstrate the effect of meta-knowledge, as more fidelity becomes available. SH similarly acts as a myopic baseline, but its schedule's path dependency may deny it to recover from early mistakes and hence result in sub-optimal states even when full fidelity is available. The steep decay in regret of MASIF at the beginning of its regret curves on Task-set and the minor adjustments on Scikit-CC18 demonstrate the desirable property, that MASIF's prior can indeed be improved based on incoming fidelity information. Similarly, IMFAS (Mohan et al., 2022) exhibits,

albeit far less pronounced, such a decay indicating that fidelity information also helps it. Its dependence on classical dataset meta-features as initialization and its autoregressive nature prevent it from excelling at increasingly revealed fidelity information and likely even is the cause of its slight deterioration on LCBench. Its strong dependence on classical dataset meta-features also prevents it from being runnable on Synthetic and Task-set.

A direct comparison of MASIF with the parametric learning curve baseline on Synthetic and Task-set in terms of reduced variability and consistent superiority – already – in the early stages highlights the benefit of having meta-learned the algorithms' learning behavior to combat myopia effectively. The SH Scheduler observed on Task-set supports this fact. When comparing SH against SH Scheduler on this noisy dataset it is apparent, that the gained meta-knowledge becomes the main driver of the regret curve. Notably, the SH Scheduler's curve appears inferior at first to MASIF's trajectory, until we factor in that MASIF follows the slice evaluation protocol very closely and the SH Scheduler has considerably less overall budget available due to the SH's budget allocation strategy. It also suffers from its path dependency, yielding slightly inferior results to MASIF, but at considerably less cost. This regret gap diminishes with lower benchmark difficulty in terms of accuracy of the meta-knowledge, as can be observed on Synthetic and Scikit-CC18, when strong prior beliefs facilitate making good decisions early on. This most prominently demonstrates the budget-regret trade-off a scheduler incurs. A comparison of SATzilla11 and our variants on Scikit-CC18 similarly highlights this trade-off; SATzilla11 incurs zero additional cost at test time over the computation of dataset meta-features which are available to all three. Its prediction must fully rely on its prior, but MASIF can reduce SATzilla11's regret at a small additional cost during test time, having learned the algorithms' learning behaviors.

To our surprise, fidelity-enabled SATzilla11 performs similarly to that without fidelity information on Scikit-CC18. This fidelity-aware but non-parametric baseline ignores the naturally occurring sequential nature and fails to improve over its fidelity-agnostic variant. It remains unclear as to why this occurs in our experiments, but we conjecture that it grossly overfits the meta-training data due to its sizable feature space.

As the ablation results in Appendix A.4 suggest, the usefulness of the raw dataset & algorithm meta-features in the two datasets is negligible at best. This has two implications: first, the meta-learned multi-fidelity information is the definite driver of our model and well outperforms the classical meta-feature-based approaches. A small caveat to the results of classical methods is their handcrafted nature. Considerable time investment into the generation and preprocessing of task-specific meta-features may yield improved regret performances for all models accepting dataset meta-features. This is why, despite the lack of strong evidence for its improvement, we keep the dataset- and algorithm meta-features as an optional part of our architecture.

## 5 Related Work

Classical AS approaches are based on the assumption that dataset instances (tasks) can be represented using meta-features. They learn mappings between datasets and algorithms w.r.t. performance based on such pre-computed meta-features. Most methods to learn such mappings in the literature either leverage regression techniques (Xu et al., 2008; Bischl et al., 2016), ranking techniques (Cunha et al., 2018; Abdulrahman et al., 2018), collaborative filtering (Stern et al., 2010; Fusi et al., 2018), instance-based learning approaches (Amadini et al., 2014; Kadioglu et al., 2010) or mixtures of the aforementioned techniques (Hanselle et al., 2020; Fehring et al., 2022). All of these approaches, however, suffer from the same limitation: Pre-computed meta-features, as mentioned earlier, can be in sufficient in characterizing the datasets and algorithms due to the alignment problem discussed in Section 2. Our approach circumvents the alignment issue by landmarking datasets directly based on the candidate algorithms' inductive biases, as unrolled in their partial learning curves, and predicts a rank over the set of candidate algorithms.

The idea of exploiting partial learning curves has previously been explored in some works. SAM (Leite & Brazdil, 2004) non-parametrically matched partial learning curves to the closest in terms of shape from the observed meta-learning curves using kNN, while van Rijn et al. (2015) reduced the cost of cross-validation by exploiting the similarity of the partially-observed rankings with those of the meta-datasets and using the most similar learning curve as surrogates. Mohr & van Rijn (2021) extended this approach by terminating less promising candidates early on based on their predicted learning curves modelled in a semi-parametric way under the assumption of concavity. In contrast to this explicit prediction of the missing parts of the learning

curves, MASIF *implicitly* meta-learns correlations between the partial performances of algorithms to predict a ranking. Leite & Brazdil (2010) pursue a cheaper and meta-informed alternative to cross-validation that evaluates the collection of fidelities using SAM, and then actively schedules them in a cost-aware manner.

Similarly, multi-fidelity optimization can be used in hyperparameter optimization (Bischl et al., 2021a) to save computing resources. It allows the optimizer to actively query new candidate configurations, unlike the classical AS setup. Corresponding methods can be categorized into bandit-based ones (e.g. SH (Jamieson & Talwalkar, 2016), ASHA (Li et al., 2020), PASHA (Bohdal et al., 2023), HB (Li et al., 2018), DEHB (Awad et al., 2021), BOHB (Falkner et al., 2018), ID-HB (Brandt et al., 2023), SMAC3 (Lindauer et al., 2022), DSMAC (Sui & Yu, 2020), BOCA (Kandasamy et al., 2017)), cost-aware ones (e.g. BOIL (Nguyen et al., 2020), FaBOLAS (Klein et al., 2017a)), and learning curve extrapolating ones (e.g. Freeze-Thaw (Swersky et al., 2014), Speeding up Automatic Hyperparameter Optimization of Deep Neural Networks (Domhan et al., 2015), DyHPO (Wistuba et al., 2022), LCNet (Klein et al., 2017c), as well as Multi-LCNet (Jawed et al., 2021)).

As Multi-LCNet is the one closest related to ours in terms of its setting on the hyperparameter optimization side, it is worth investigating the differences more closely. While they use a related $\Omega^{meta-train}$ and $\Omega^{meta-test}$ data setup, the work has substantial differences and rigid assumptions, which make us abstain from an experimental comparison of MASIF to it, as a fair comparison is not possible. The core differences are threefold; 1. they consider a hyperparameter optimization multi-fidelity problem 2. their model can assume the existence of auxiliary curves for a single algorithm, because of the narrow definition of an algorithm and 3. they use this multi-modal data in an autoregressive way for the explicit extrapolation of a single curve at a time. A more detailed discussion is deferred to Appendix A.5.

In summary, all of these methods are applications of multi-fidelity in hyperparameter optimization. From the lens of our setup, they are applied to a relaxation of the hyperparameter search space that is out of scope here.[5] Crucially the first two kinds of methods primarily focus on the problem of scheduling the configuration and endowing it with a budget. On the other hand, the last kind of method focuses on marginally extrapolating learning curves for a single configuration irrespective of previously seen learning curves on the same dataset. MASIF differs from these approaches in that it focuses on jointly interpreting the learning curves in the AS setup and does not imply any schedule.

Two approaches orthogonal to ours, although bearing some resemblances in terms of setup, are Meta-REVEAL (Nguyen et al., 2021) and MetaBu (Rakotoarison et al., 2022). Meta-REVEAL focuses on scheduling the algorithms and fidelities through a Reinforcement Learning perspective by modelling it as a REVEAL game. The agent acts on a discrete action space that does not account for the correlations between learning curves. As we do not focus on schedule but on learning cross-correlations on the learning behavior, we refrain from a comparison. Consequently, using the estimates produced by MASIF can provide a pre-processed action space for their problem. MetaBu extends the classical AS idea by relating dataset meta-features and algorithms' hyperparameters using a learned optimal transport map. They, however, ignore that the dataset meta-features cannot sufficiently characterize the dataset, thus, subsequently falling victim to the same fallacies as the methods mentioned before.

## 6 Conclusion

In this work, we revisited classical meta-learning- and learning curve-based approaches to AS from the perspective of a trade-off between the budget they invest in inquiring information about a new dataset and the corresponding regret of choosing an inferior algorithm. Doing so, we argued that both of these selector classes are on opposite ends of the spectrum in this trade-off and both suffer from severe limitations, likely leaving data scientists with a selected algorithm that is inferior. To alleviate this, we present MASIF, an algorithm selector designed to support data scientists in the selection process in a far more native manner. Its position in the spectrum can be controlled by the data scientist depending on their budgetary constraints. MASIF leverages a transformer-encoder-based architecture to model the performance of algorithms across both, fidelities and algorithms in the form of learning curves of varying lengths. As such, it alleviates the myopia

---

[5]SH-based methods are living on a discretized version of this broader space.

of many existing learning curve-based approaches while making no parametric assumptions on the learning curves it can model. Crucially, our selector utilizes computation from previous selections as meta-knowledge. In an extensive experimental study on four different benchmarks, we showed that MASIF outperforms existing meta-learning-based approaches in terms of the regret of the selected algorithm and learning curve-based algorithm selectors in terms of regret for the invested budget. As such, MASIF is not only an AS approach designed to support the data scientist in practice by leaving the concrete instantiation of the trade-off to their preference but also yields state-of-the-art AS performance.

## 7 Future Work

The aforementioned budget-regret trade-off leads to a multi-objective view of the AS problem, where both the budget invested and the regret of the corresponding algorithm selector are rivalling objectives. This holistic view naturally suggests tackling the problem with multi-objective methods to be able to present a Pareto front of selectors to choose from, suited to a data scientist's budget-regret preferences in the face of uncertainty. This line of work is orthogonal to previous multi-objective work in Algorithm Selection (Bossek & Trautmann, 2012). Any instance on this Pareto front is a scheduler of sorts. Determining how a scheduler knowledgeable in learning behaviors should choose its budget allocation sequentially and in the face of uncertainty inspires research on its own. Lastly, in the current form of this work, the algorithms are predominantly represented through their learning curves and we use algorithm meta-features in a relatively limited manner. Multi-LCNet (Jawed et al., 2021) presumes that using hyperparameters among other informative algorithm meta-features may also be beneficial in contextualizing a learning curve. In that sense, a natural extension to the scope of this work is the transition from tackling the AS problem to tackling the hyperparameter optimization problem. In contrast to existing approaches, we will seek to extract the joint topology evidence derived from the observed partial learning curves in order to tackle this problem.

### Acknowledgements

Marius Lindauer and Tim Ruhkopf acknowledge financial support from the Federal Ministry for Economic Affairs and Energy of Germany in the project CoyPu under Grant No. 01MK21007L.

Frank Hutter acknowledges financial support by the European Union (via ERC Consolidator Grant "DeepLearning 2.0", grant no. 101045765). Alexander Tornede acknowledges similar financial support by the European Union (via ERC, "ixAutoML", grant no.101041029). Views and opinions expressed are however those of the authors only and do not necessarily reflect those of the European Union or the European Research Council. Neither the European Union nor the granting authority can be held responsible for them.

The authors gratefully acknowledge the computing time provided to them on the high-performance computers Noctua2 at the NHR Center PC2 under the project hpc-prf-intexml. These are funded by the Federal Ministry of Education and Research and the state governments participating on the basis of the resolutions of the GWK for the national high performance computing at universities (www.nhr-verein.de/unsere-partner).

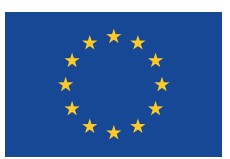
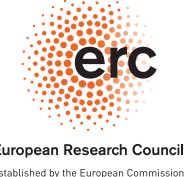

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

# A  Appendix

## A.1  Meta-Datasets

**Synthetic**   To demonstrate that MASIF is a non-myopic approach, we create a synthetic function meta-dataset. A snapshot of the synthetic curve meta-dataset can be found under Figure 4. To break the myopic algorithms, we force the curves in one dataset to intersect at least another curve in an incrementally way. We initialize by randomly picking a parametric curve. Subsequent curves are generated by first sampling a parametric family as described by Mohr et al. (2022) and then computing its parametrization such that it is ensured to intersect with its predecessor. In more detail, we want to ensure that the intersections occur at different stages of the training procedure. This is obtained by

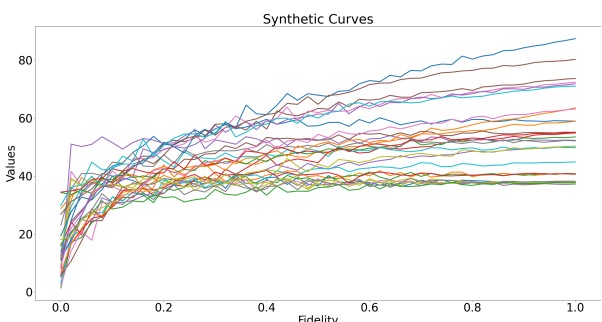

Figure 4: A snapshot of a Synthetic dataset

randomly selecting one of a few preset intervals in which the intersection is supposed to take place. Repeating this process for $|\mathcal{A}| - 1$ times produces a single dataset. To make the task more challenging, we add random noises to the generated curves. To obtain strong meta-knowledge across datasets originating from the shape of an algorithm's curve, a new dataset is generated by perturbing the parametrization of the existing curves. Therefore, these datasets contain similar but non-identical curves, that yield variability in the final ranking. This dataset does not provide dataset meta-features. Overall, we collect 22 datasets each containing 30 learning curves with 51 fidelities.

**Task-set**   Task-set (Metz et al., 2020) is a meta-dataset that consists of over a thousand tasks ranging from image classification with fully connected or convolutional neural networks to variational autoencoders on a variety of datasets. Each task is characterized by 1. an initialization function 2. data split 3. loss function 4. gradients

On these tasks, the training, validation, and test curves of multiple optimizer settings have been recorded. The tasks are additionally organized into families, each pertaining to the kind of problems that they are trained on. For our experiments, we sub-sampled the tasks from 2 different categories of tasks: Image Recognition and Language Modeling on text data.

The image recognition subset comprises curves generated from fully-connected networks run only on image recognition tasks, encompassing the following families:

1. **mlp:** fully-connected networks trained on image data

2. **mlp_ae:** MLP-based autoencoder trained on image data

3. **mlp_vae:** MLP-based VAE trained on image data

The language modelling subset comprises curves generated from Recurrent Neural Networks (RNNs) trained on text data, encompassing the following three families:

1. **char_rnn_language_model:** Language modelling with an RNN on characters.

2. **word_rnn_language_model:** Language modelling with an RNN on words and subwords.

3. **rnn_text_classification:** n Text classification using RNN models.

We sample the set of algorithms to be the different configurations of the Adam optimizer (Kingma & Ba, 2015) with variations of the learning rate. This allows us to create a meta-dataset of 1000 configurations on

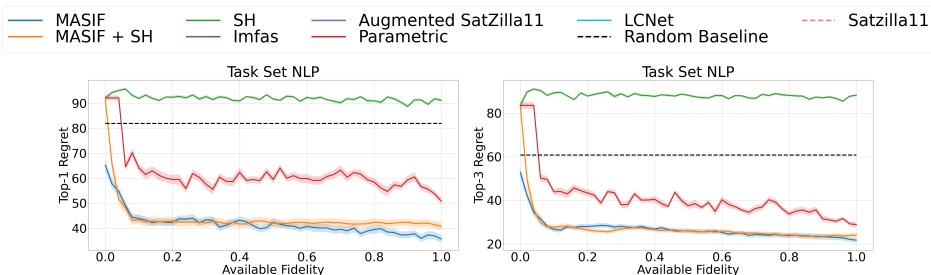

Figure 5: Average test-set top-1 and top-3 regret over ten meta-dataset folds in the slice evaluation protocol for the NLP subset of Task-set. Available fidelity is expressed in the share of the target fidelity's budget. The standard deviation originates from five repetitions for each fold.

100 tasks. This meta-dataset does not provide dataset meta-features. The rationale is that given the noise and likely crossings of the curves sampled from realistic tasks, the performance on initial fidelities is not indicative of the final ranks, making it hard for methods like Successive Halving to get a good regret value without any form of meta-knowledge.

**Scikit-CC18** OpenML-CC18 (Bischl et al., 2021b) is a curated classification benchmark featuring 72 carefully selected datasets from OpenML (Vanschoren et al., 2014) with a variety of desirable properties for a benchmark (see (Bischl et al., 2021b) for details). We generated learning curves on these datasets for 16 classifiers from Scikit-learn (Pedregosa et al., 2011), namely (i) ExtraTreeClassifier (ii) DecisionTreeClassifier (iii) MLPClassifier (iv) KNeighborsClassifier (v) SGDClassifier (vi) RidgeClassifier (vii) PassiveAggressiveClassifier (viii) GradientBoostingClassifier (ix) ExtraTreesClassifier (x) BernoulliNB (xi) LinearSVC (xii) LogisticRegression (xiii) MultinomialNB (xiv) NearestCentroid (xv) Perceptron (xvi) SVC.

We collect their learning curves performing the following procedure for every dataset and learner pair: We performed three-fold cross-validation and for each of these folds again split 20% off of the training data as validation data. We then trained the classifier on $5\%, 10\%, \ldots, 100\%$ of the training data and evaluated it on the train, test, and validation data using accuracy as a loss function. As preprocessing, we imputed missing feature values with the most frequent one in the training data and one-hot encoded categorical features. Correspondingly, the budgeted resource is the dataset subset size for this meta-dataset. This dataset provides dataset meta-features.

**LCBench** LCBench (Zimmer et al., 2021) is a meta-dataset, that consists of 2000 hyperparameter configurations of a funnel-shaped neural net on 35 datasets. To meet the AS setup, we choose a subset of size 170 of these configurations using a top-$k$ ensembling procedure. In particular, this procedure builds the union of algorithms over the top-10 performing algorithms per dataset across datasets to ensure that each algorithm was at least somewhat successful on at least one dataset. Presuming, that hyperparameters change the inductive bias of a model, we conceive for the purpose of our analysis these configurations as independent algorithms. Notably because of this, the hyperparameter optimization usually resorts to assuming similarity of performance w.r.t. hyperparameters. By extension, we expect that we can assume some similarity in the learning behaviors encoded in their learning curves, that MASIF should be able to exploit. This meta-dataset provides dataset meta-features.

## A.2 Task-set NLP

On the NLP task, depicted in Figure 5, we see the overall tendencies of the image subset Task-set, with the exception that the parametric learning curve baseline is not able to close the performance gap provided full fidelity.

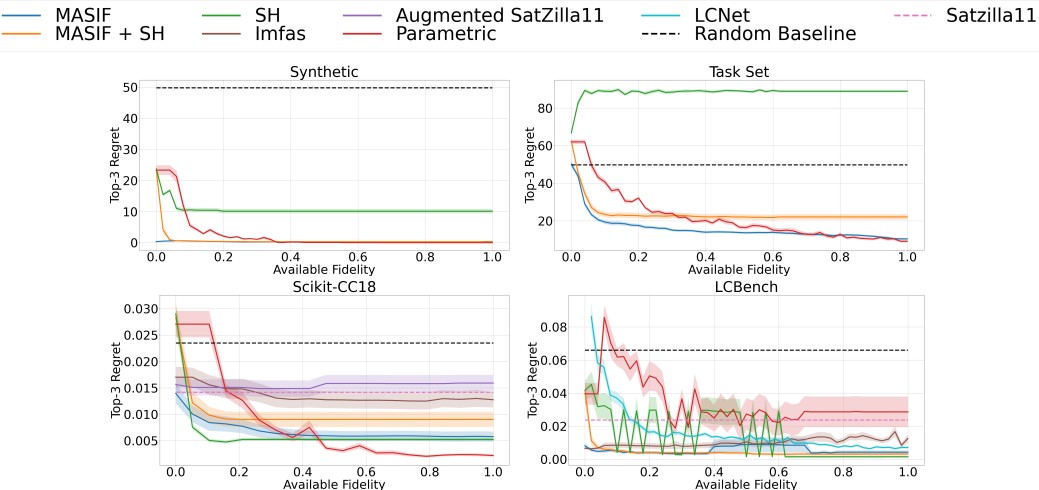

Figure 6: Average test-set top-3 regret over ten meta-dataset folds in the slice evaluation protocol. Available fidelity is expressed in the share of the target fidelity's budget. The standard deviation originates from five repetitions for each fold. Scikit-CC18 and LCBench exhibit dataset meta-features, while Synthetic and Task-set do not. Task-set is reported for the image data subset.

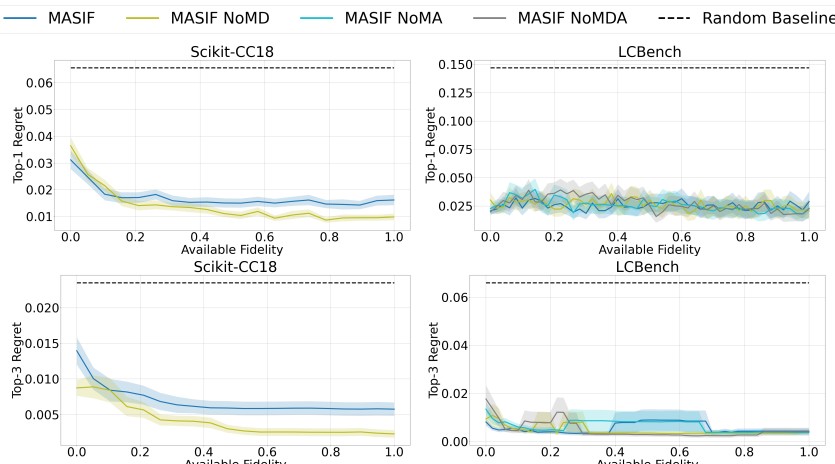

Figure 7: Ablation of MASIF w.r.t. whether or not meta-features are available on Scikit-CC18 (Left) and LCBench (Right) benchmarks. 'NoM' indicates that the subsequently detailed meta-feature information is hidden to the model. In particular, 'D' stands for dataset meta-features, 'A' for algorithm meta-features.

## A.3 Top-3 results

As increasing $k$ in top-$k$ is a hedge, the overall regret is reduced compared to those results of top-1, because it is more likely to pick the best performer in this enlarged set. The overall tendencies in Figure 6 are however exactly the same as described in Section 4.4.

## A.4 Ablations

### A.4.1 Dataset & Algorithm Meta Features

The results in Figure 7 indicate that the regret difference in whether or not the meta-features are present have – if any – only negligible effect for the Scikit-CC18 and LCBench dataset, once the scale of this difference is considered. On the other hand, the results in Figure 8 detail, the regret gap, when using a secondary

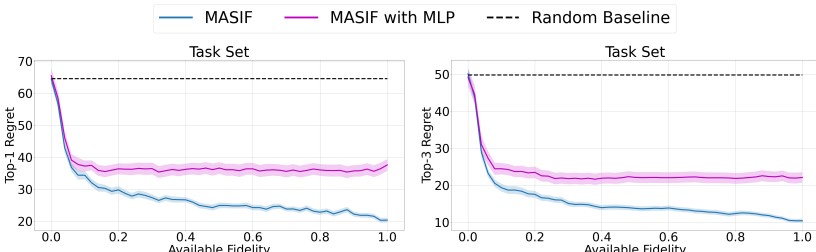

Figure 8: Ablation of MASIF on Task-set benchmark, switching out the second transformer for a simple MLP on the reduced output as joint interpretation module. Task-set is reported for the image data subset.

transformer over the sequence of algorithms rather than an MLP, supporting our design choice for the former. The vector representation of the learning curves as output to the first transformer holds positional information since an algorithm is presented always in the same order irrespective of the dataset. The secondary transformer picks up on this positional information more easily than an MLP.

### A.4.2 Parametrization

Provided the scarce data available, we cannot reasonably perform hyperparameter optimization on the meta-level without risking overfitting. Instead, we consider how the design choices regarding the model's hyperparameters are affecting performance marginally on Task-set, depicted in Figure 9. As AdamW with 5% linear warm-up and cosine decay is a default choice for optimizing transformers, we added an ablation to it. Since the number of attention heads, the number of transformer layers affect the model's capability in describing functions, these hyperparameters are relevant choices. Similarly, dropout should help the robustness of the transformer. We do, however, find that our model is relatively robust to changes in these hyperparameters. The overall tendencies and results are the same across hyperparameter configurations. The slight performance improvements for the marginal alterations prompted us towards trying their combination, i.e. AdamW with linear warm-up and cosine decay with a learning rate of 0.001, dropout of 0.4 and 4 as number of transformer layers (as depicted in Figure 9 bottom right). To validate the design choice, we checked the performance of this promising combination on the remaining benchmarks as depicted in Figure 10 as a validation set. We do not find the improvements observed on Task-set to consistently yield improvements on these benchmarks, which is why we are defaulting to the hyperparametrization described in Section 3.4.

### A.5 Discussion on Multi-LCNet

As it is conceptually the closest to our model, we will discuss the differences summarized in 5 more thoroughly. First, their method considers hyperparameters, which allows exploiting the similarities in that space to interpolate between algorithms - which eases our assumption of a discrete and rigidly observed $\mathcal{A}$ in the meta-training set and therefore is rather hyperparameter optimization than AS. Second, they leverage multi-modal data; i.e. multiple auxiliary curves such as e.g. the layer's gradient information are tracked during training. These curves are only accessible for all configurations, because they limit themselves to a single class of algorithms, in their case funnel-shaped neural nets. These secondary curves are not necessarily available for all candidate algorithms considered in an AS model and as such, the method cannot be applied in our setting. Besides, considering SH's almost oracle performance on LCBench in Figure 3 with a sizeable subset of 170 algorithm configurations in our experiments, the multi-modal nature of their analysis may not be a driver for their performance on this particular dataset. Third, their focus is on explicitly extrapolating single learning curves at a higher fidelity in an auto-regressive manner based on the meta-learned weights of their Gated Recurrent Units (Cho et al., 2014) that only combine the information from the partial learning curves of the same algorithm observed from multiple modes of data. Instead, our method leverages the joint evidence of all invested computations in addition to the meta-knowledge and avoids explicit extrapolation. Fourth, their handcrafted loss metric is intended to foster good predictions early on, which, despite working with rather small budgets, might yield suboptimal decisions. Similarly to the SH baseline, this seems to

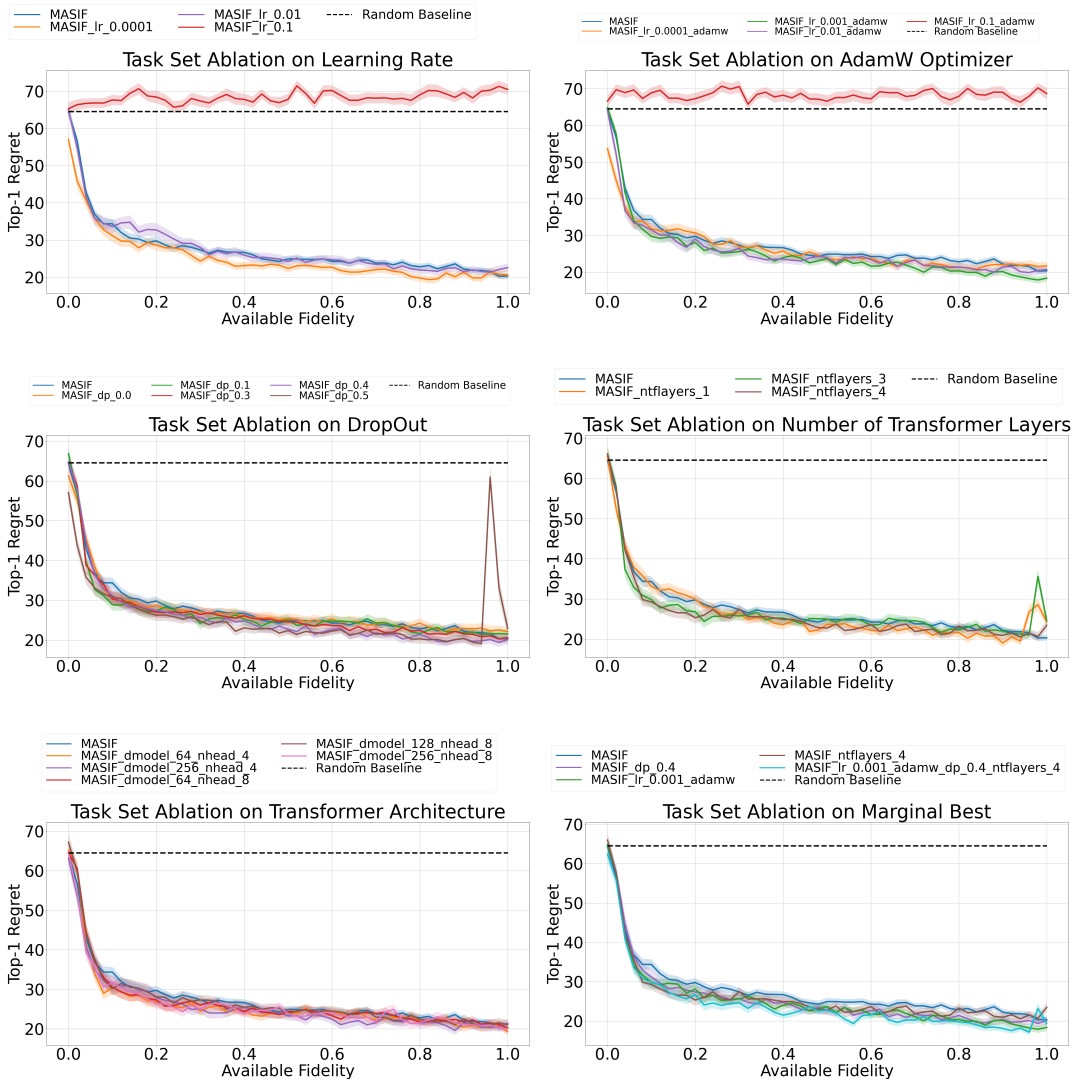

Figure 9: Ablations of MASIF with marginal changes in the hyperparametrization on Task-set benchmark on all its folds with 5 seeds respectively. The marginally changed parameters are Adam's learning rate (lr), the learning rate of AdamW with 5% linear warm-up and cosine decay (lr_adamw), dropout (dp), number of attention heads (n_head), the number of transformer layers (nftlayers) and their hidden dimensions (d_model). Notably, we jointly looked at d_model & n_heads. The default configuration is depicted in blue and amounts to lr=0.001, dp=0.2, n_heads=4, ntflayers=2, d_model=128. Task-set is reported for the image data subset.

express and favour a particular budgetary preference over the resulting regret. In contrast, as mentioned several times, MASIF is much more flexible regarding the budget-regret trade-off.

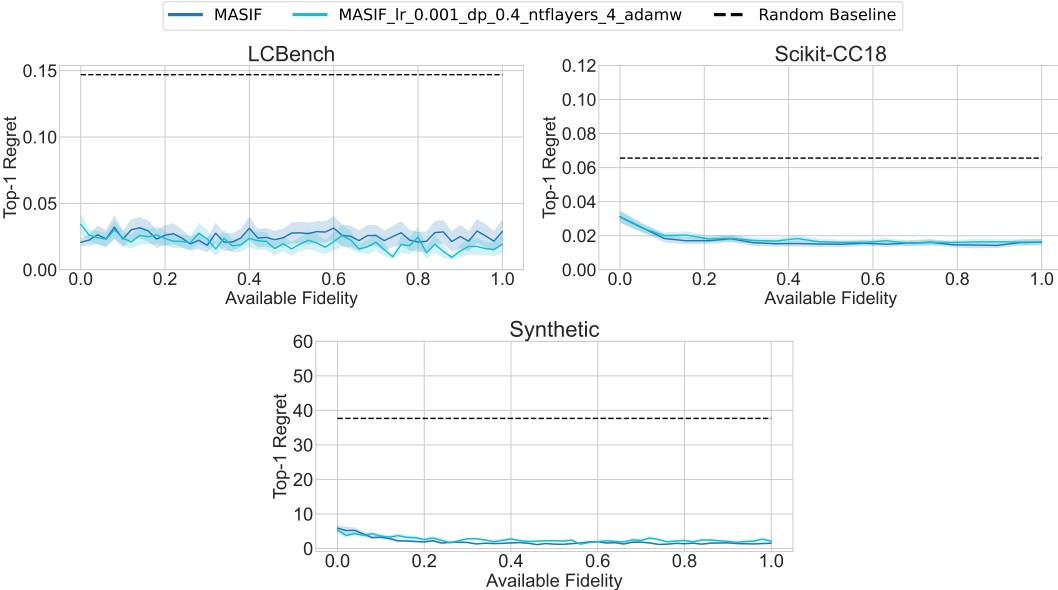

Figure 10: We apply the optimal configuration on the Taskset benchmarks to the other benchmarks. However, the optimal configuration on the Taskset benchmark does not show better performance compared to MASIF's default setting.

**A.6 Reproducibility Statement**

- Where can the code be found? MASIF's code is published on `https://github.com/automl/masif`.

- What hardware did we use for the experiments? All the experiments are executed on 4 Intel Xeon E5 cores with 8000MB RAM.

- What hyperparameter settings did we use? How did we get to those? Where can they be found? Using Hydra (`hydra.cc`) as base package for all our experiment pipelines – including the preprocessing of dataset & algorithm meta-features as well as those for the learning curves, all of the configurations detailing the used models and their configurations can be found in the "configs" folder in the linked repository.

- What are the requirements in terms of packages and version numbers? The packages & version numbers are available in the setups file of the linked repository.

- Where can you download our benchmark datasets? We added the newly created Scikit-CC18 benchmark as supplementary material. LCBench can be downloaded from `https://github.com/automl/LCBench`. Task-set can be obtained from `https://github.com/google-research/google-research/tree/master/task_set`. We provide the Synthetic and Scikit-CC18 benchmarks as supplementary.

- How many seeds/splits etc. did we perform? We used five seeds and ten folds for each benchmark. A crucial detail regarding the split in the algorithms' learning curves is described in the dataset section.

