# OpenReview forum: "MASIF: Meta-learned Algorithm Selection using Implicit Fidelity Information"
_TMLR — Accepted by TMLR_

### Review · Reviewer_p2g7 · 2022-12-18

**Summary Of Contributions:**

The authors use a Transformer model they call MASIF to rank machine learning training algorithms by how effective they are predicted to be on a new task. They validate this setup on several datasets of machine learning tasks.

**Audience:**

Yes

**Claims And Evidence:**

Yes

**Requested Changes:**

Was “guided attention” necessary to get good performance, or could you have used a standard architecture? Using a more common model could be useful because you can more easily take advantage of optimized implementations.

I find it suspicious that you moved the experiments where your method does not perform as competitively to the appendix (A.3). I don’t think it’s a requirement that a proposed method be SOTA on all experiments run, so I think this should go in the main text.

Nothing about how you actually trained the MASIF model was included in the paper, information which should absolutely be present for any paper doing deep learning experiments. Eventually I found https://anonymous.4open.science/r/MASIF-824D/configs/trainer/basetrainer.yaml which shows Adam with an LR=1e-3. Did you tune this at all? I think it’s very fair to try different settings of your algo selector hparams to try to get better performance. Also are you using a learning rate schedule for training MASIF? If not, I _highly_ recommend using a linear warmup for 5% of training followed by a cosine decay, as these seem to be universally useful for Transformer training (at least compared to a constant LR).

Looking at your model configs https://anonymous.4open.science/r/MASIF-824D/configs/model/masif_m_transformer.yaml I see you use a dropout=0.2. Did you tune this? Typically 0.1 is used, and that may give better performance, and unless you’re really worried about overfitting you could even turn this off entirely.

How did you pick the model sizes? If you are training on all 29 million task set curves (https://github.com/google-research/google-research/tree/master/task_set), then you may be able to get performance gains by increasing the model size beyond the 2-3 128/256/512 dimensional layers I see in the configs https://anonymous.4open.science/r/MASIF-824D/configs/model/masif_transformer.yaml.

**Strengths And Weaknesses:**

Paper is well-written but verbose. For example, the MASIF architecture description is lengthy and could have gone in the appendix, and presumably the reader does not need to be told things such as “The attention guided transformer-encoder layer returns a tensor of the same dimension as its input. This layer is repeated N times.” Section 3.3 could likely be a single sentence. However, information that I would expect to be in the model description is missing, such as the actual sizing of the layers and weights used (these can be found if you dig into the codebase, but a brief summary of the model size would be very useful in the paper).

Overall I think the paper is fine enough for acceptance.

---

> ### Author Response · Authors · 2023-01-13
> **First revision**
>
> Thank you very much for your detailed and particularly technical review as well as your welcomed suggestions. We have summarized some of your changes in the general comments to all reviewers.
> > Was “guided attention” necessary to get good performance, or could you have used a standard architecture? Using a more common model could be useful because you can more easily take advantage of optimized implementations.
>
> Regarding your request as to whether guided attention was necessary to get good performance, we would like to point you to the discussion in the last paragraph of Section “4.4 Results / The role of meta-knowledge”, summarizing the ablation in Appendix. 5 Ablations. As our data setup is novel, we are only aware of LCBench as a multi-fidelity benchmark with full learning curves and algorithm and dataset meta-features. We also considered the following benchmarks which all had different limitations:
>
> (i) [Pfisterer et al., YAHPO Gym - An Efficient Multi-Objective Multi-Fidelity Benchmark for Hyperparameter Optimization (PMLR2022)](https://proceedings.mlr.press/v188/pfisterer22a.html).
> While it has dataset meta-features and algorithm meta-features, it lacks the necessary test data for a proper meta-test split. It offers only validation and training curves, but not the test curves. This could cause biased results which is why we decided against it.
>
> (ii) [Mohr et al., LCDB 1.0: An Extensive Learning Curves Database for Classification Tasks (ECML 2022)](https://2022.ecmlpkdd.org/wp-content/uploads/2022/09/sub_1317.pdf). LCDB, while extensive in the number of experiments, it primarily was conceived to validate a hypothesis on the individual learning curves. As a consequence, the resulting data is hard to aggregate into a learning curve tensor. In particular, the spacings of the fidelities are irregular between algorithm executions across datasets. We had considerable issues with finding a viable subset of algorithm-dataset-fidelity combinations that is sufficient for our purposes.
>
> (iii) [Eggensperger et al., HPOBench: A Collection of Reproducible Multi-Fidelity Benchmark Problems for HPO (NeurIPS 2021)](https://datasets-benchmarks-proceedings.neurips.cc/paper/2021/hash/93db85ed909c13838ff95ccfa94cebd9-Abstract-round2.html) When we assessed HPOBench’s multi-fidelity data, it either only exhibited the fidelity levels 0.1, 0.33 and 1.0 or only the final fidelity, which the authors confirmed. It was insufficient for our purposes.
>
> The limited empirical evidence on LCBench (now in Section 4.4) neither provides a strong signal for one or the other. However, from a theoretical point of view, we believe that guided attention is reasonable because, as outlined in Block 3 of our architecture, the complexity of a dataset might already be somewhat captured in the dataset meta-features and alter the shape of the learning curve - hence knowledge about such a change is beneficial.
>
>
> > Paper is well-written but verbose. For example, the MASIF architecture description is lengthy and could have gone in the appendix, and presumably, the reader does not need to be told things such as “The attention-guided transformer-encoder layer returns a tensor of the same dimension as its input. This layer is repeated N times.” Section 3.3 could likely be a single sentence. However, information that I would expect to be in the model description is missing, such as the actual sizing of the layers and weights used (these can be found if you dig into the codebase, but a brief summary of the model size would be very useful in the paper).
> Nothing about how you actually trained the MASIF model was included in the paper, information which should absolutely be present for any paper doing deep learning experiments. Eventually, I found https://anonymous.4open.science/r/MASIF-824D/configs/trainer/basetrainer.yaml
>
> Regarding the verbosity of Section 3.2 (MASIF Architecture), we conceive these details as essential information for reproducing and re-implementing our approach. Thus, we prefer to keep them in the main paper. We hope that a knowledgeable reader will simply skip that part.
>
> Thank you for pointing out the missing information regarding the model size and training procedure. We have added the information in Section 3.4 “Hyperparameters”.

---

> > ### Author Response · Authors · 2023-01-13
> > **Continuation: First revision**
> >
> >
> > > which shows Adam with an LR=1e-3. Did you tune this at all? I think it’s very fair to try different settings of your algo selector hparams to try to get better performance. Also, are you using a learning rate schedule for training MASIF? If not, I highly recommend using a linear warmup for 5% of training followed by a cosine decay, as these seem to be universally useful for Transformer training (at least compared to a constant LR). Looking at your model configs https://anonymous.4open.science/r/MASIF-824D/configs/model/masif_m_transformer.yaml I see you use a dropout=0.2. Did you tune this? Typically 0.1 is used, and that may give better performance, and unless you’re really worried about overfitting you could even turn this off entirely. How did you pick the model sizes?
> >
> > Thanks for pointing out how we can further improve our model. We did a couple of experiments with your proposed settings to shed light on the importance of these design decisions and hyperparameter settings of our meta-model. We see minor improvements on Taskset by using these compared to our default settings. However, we note that simply tuning these hyperparameters on the final benchmarks can potentially lead to overtuning our model to an individual benchmark; in fact, using the same tuned settings on the other benchmarks has not improved their results, indicating overfitting of the architecture and hyperparameters to the Taskset benchmark. Due to the sparsity of data we are obstructed from doing a proper meta-level hyperparameter optimization.
> > As MASIF is composed of two subsequent transformer encoders that apply attention operation to different levels, we wanted to start with a relatively small model given the limited amount of algorithms and datasets in the benchmarks. As the ablation study in Appendix A.4.2. reveals, the architecture is relatively robust against changes in these design choices.
> >
> >
> > > Was “guided attention” necessary to get good performance, or could you have used a standard architecture? Using a more common model could be useful because you can more easily take advantage of optimized implementations.
> >
> > See our answer above to that question.
> >
> >
> > > If you are training on all 29 million task set curves (https://github.com/google-research/google-research/tree/master/task_set), then you may be able to get performance gains by increasing the model size beyond the 2-3 128/256/512-dimensional layers I see in the configs https://anonymous.4open.science/r/MASIF-824D/configs/model/masif_transformer.yaml.
> >
> > We agree with this hypothesis. Nevertheless, we decided to run our benchmark on a considerably smaller subset of learning curves from Task-set as outlined in the appendix. This is due to the fact that we try to deal with an AS problem, rather than an HPO problem — and our architecture needs to take in all of the “algorithms” learning curves on a task at once, to jointly interpret their evidence. It will be part of future work to extend the ideas of MASIF beyond AS and scale it up to HPO, which will most likely also include more layers.

---

> > > ### Comment · Reviewer_p2g7 · 2023-01-30
> > > **ack**
> > >
> > > Thanks for the detailed responses, these answered my questions!

---

### Review · Reviewer_UzPx · 2022-12-19

**Summary Of Contributions:**

This paper proposes a new algorithm for Algorithm Selection (AS) where the goal is to predict the relative performance of different learning algorithms without the need to execute the whole algorithm fully. This process usually involves using cheap-to-compute features of the data and learning algorithms that contain information about the final performance of the algorithms. The proposed framework MASIF (Meta-learned Algorithm Selection using Implicit Fidelity) additionally leverages the training information from different learning algorithms to meta-learn relationships between different features and the final performance of the model. MASIF leverages a specially designed transformer architecture that takes in information about the data, algorithm hyperparameters and values of the training loss curve and predicts a score that is used to rank different learning algorithms. The architecture takes into a set of learning algorithm features and outputs each algorithm’s score and a ranking loss (between the ranking of predicted scores and true ranking) is reduced via SGD. Experiments show that MASIF outperforms various baselines on 3 benchmarks.

**Audience:**

Yes

**Broader Impact Concerns:**

The paper studies the general problem of algorithm selection, so there is no obvious ethical concern as far as I can tell. On the other hand, it could inherit most standard problems in data-driven machine learning such as amplifying bias in the data, etc.

**Claims And Evidence:**

Yes

**Requested Changes:**

- The definition of $\mathcal{D}$ is never explicitly specified.
- Equations 3, and 4 are wrong or at least inconsistent. $s(D)$ outputs an algorithm, but $\mathcal{L}$ takes in a ranking.
- Equation 3 also never specifies what the loss for ranking is and I can't see to find this information in the appendix either.
- Line 84 $m$ is different from line 138 $m$ (i.e., they have different arguments).
- Elaborate why the algorithm features are treated as position encodings or have an appropriate ablation study
- Have more ablation studies about different components of the transformer architecture


**Strengths And Weaknesses:**

**Strength**
- The proposed algorithm brings a new perspective to the problem of AS
- The usage of learning curves for AS is novel but very intuitive
- The improvement of the proposed algorithm over the existing method is significant.

**Weakness**
- The paper’s writing quality can be improved. It has many terminologies which are already hard to keep track of and sometimes the notations are inconsistent, making the paper really hard to read. It would benefit from some cleaning up.
- The proposed architecture is quite complicated and there are not sufficient ablation studies to demonstrate the effects of each component of the architecture. For example, for block 4 of figure 2, why is the algo feature embedding used as a positional encoding rather than the more conventional one-hot encoding? Another example is the usage of guided attention, which is also never ablated. These ablation studies are quite important for algorithms as complicated as this as we can see in the appendix that the utility of block 1 (meta-features) is actually unclear so it would be good to know what each part does.
- See the Requested changes for specifics.

---

> ### Author Response · Authors · 2023-01-13
> **First revision**
>
> Thank you very much for your review and particularly your sharp theoretical observations. Please consider the general remarks to all the reviewers. Other than that, we hope to have addressed your concerns with the following changes:
>
>
> > The proposed architecture is quite complicated and there are not sufficient ablation studies to demonstrate the effects of each component of the architecture. For example, for block 4 of figure 2, why is the algo feature embedding used as a positional encoding rather than the more conventional one-hot encoding? Another example is the usage of guided attention, which is also never ablated. These ablation studies are quite important for algorithms as complicated as this as we can see in the appendix that the utility of block 1 (meta-features) is actually unclear so it would be good to know what each part does.
>
> We have added a rationale for positional encoding vs one-hot encoding in Block 4. Additionally, we have added an ablation to check the design choice in Block 4 in Appendix A.4.1. We note that our ablations are limited in scope due to the structure of the available benchmarks, in particular unavailable meta-features for some of them. We hope to guide future research in this direction and insprire more comprehensive benchmarks.
>
>
> >The definition of $D$ is never explicitly specified.
>
> Thanks for pointing that out. We have added a short informal definition in Line 81.
>
>
> > Equations 3, and 4 are wrong or at least inconsistent. $s(D)$ outputs an algorithm, but $\mathcal{L}$  takes in a ranking.
>
> Regarding $s(D)$’s definition we would like to point out that there is a deliberate and distinct difference between the classical setup in the preliminaries Line 81 and our new interpretation which generalizes this narrow definition in Line 105. So, with our new definition in Line 105, Equations 3 and 4 are consistent.
>
>
> > Line 84 $m$  is different from line 138 $m$  (i.e., they have different arguments).
>
> Similarly regarding the intended change in definition, we hope that with the subscript $m_{lc}$, we clarified the notation of m in the second definition.
>
>
> > Equation 3 also never specifies what the loss for ranking is and I can't see to find this information in the appendix either.
>
> Regarding the loss we used, we refer to Section 3.2 MASIF Architecture/ Block 5, the last two sentences. Since the section you are referring to is the preliminaries, where we seek to introduce the class of classical Algorithm Selectors, we are deliberately vague here. The instances of selectors can be trained on different ranking losses as for instance a differentiable version of NCDG@K or Plackett-Luce.
>
>
> > Elaborate why the algorithm features are treated as position encodings or have an appropriate ablation study
>
> We have added details to our previous description as to why we chose to consider the algorithms’ meta-features (if they are available or a simple one-hot as replacement) as positional encoding. In short: since the algorithm meta-features’ latent representation may hold information on how the learning curve is to be contextualized, we redefine the use of the positional encoding as a way of entering this information. Furthermore, as without the positional encoding, the position is lost to the transformer, but the fact that each algorithm is presented on the same position across datasets, it holds particular meaning in contextualizing the learning curve embedding. Even if the positional encoding is merely an index, detailing that the presented learning curve originates from a particular algorithm.

---

> > ### Comment · Reviewer_UzPx · 2023-02-21
> > **Thank you for the revision.**
> >
> > Thank you for the revision and clarification. My concerns and questions are mostly addressed.
> >
> > However, I would suggest the authors to avoid overloading any terms because they could cause confusion.
> >
> > > we have added an ablation to check the design choice in Block 4 in Appendix A.4.1
> >
> > I would also make sure that A.4.1 is referred to in the text.
> >
> > > where we seek to introduce the class of classical Algorithm Selectors, we are deliberately vague here
> >
> > I’m not sure about the benefits of being “deliberately vague”. I believe that the purpose of a research paper is to be as clear as possible. You could achieve the same effect by saying something like “this is what classical algorithms do but we will introduce something later”.

---

> > > ### Author Response · Authors · 2023-02-22
> > > **Latest Revision**
> > >
> > > >I would also make sure that A.4.1 is referred to in the text.
> > >
> > > With the latest revision we have added this reference at the end of Block 4.
> > >
> > > > I’m not sure about the benefits of being “deliberately vague”. I believe that the purpose of a research paper is to be as clear as possible . You could achieve the same effect by saying something like “this is what classical algorithms do but we will introduce something later”.
> > >
> > > Thank you for that suggestion. We agree that research should be as clear as possible. But since there is a plethora of approaches with different loss functions eventually aimed at creating either an implicit (e.g. MSE on the final performances as training loss to then subsequently order them by predicted performance) or explicit ranking. With our notation, we aimed at unifying these approaches. We are specific about our own version of this ranking loss.

---

### Review · Reviewer_bHsm · 2023-01-22

**Summary Of Contributions:**

This paper introduces a new meta-learning method for algorithm selection called MASIF that uses multi-fidelity learning curves from previously seen tasks and associated meta-features to learn a model for rankings across a fixed set of hyperparameters.  This model can be used for new tasks over the same hyperparameter space to predict hyperparameter rankings given partial learning curves to speed up algorithm selection.  In contrast to prior work, MASIF does not rely solely on cheap to compute landmark features to inform selection of new configurations nor make parametric assumptions about learning curve for future extrapolation. The authors evaluate MASIF on 4 different benchmarks against established baselines and show MASIF to yield competitive performance across all benchmarks.  Finally, the authors discuss how MASIF can be used in a data scientist's typical workflow to inform budget-regret decisions on which configurations to continue training.

**Audience:**

Yes

**Broader Impact Concerns:**

None.

**Claims And Evidence:**

Yes

**Requested Changes:**

- Strengthen: add better baselines like [BOHB](https://arxiv.org/abs/1807.01774) and [Learning Curve Prediction with Bayesian Neural Networks](https://openreview.net/pdf?id=S11KBYclx).  I understand that the problem solved by BOHB is different since it can sample from the search space but this is a strictly harder problem than selecting from a finite set as MASIF is doing.  I can imagine running BOHB with different fidelities and giving it the same number of tries as the finite set considered by MASIF and report the final performance of the best found configuration.
- ~~Critical: run a version of SH/BOHB that uses the same resources as that given to MASIF.  For example, MASIF sees learning curves trained on a total of # configurations * fidelity, which will be the resources allocated to SH/BOHB.~~
- ~~Critical: please check result for SH on Taskset.  SH is able to learn something useful on that benchmark according to Figure 3 of [this paper](https://arxiv.org/pdf/2202.09774.pdf).~~
- Strengthen: please evaluate MASIF on Taskset with all datasets included instead of just the selected tasks with MLP on image datasets.
- ~~Critical: please provide clarification on TaskSet search space.  The meta-dataset is described as 1000 configurations on 100 tasks in the appendix, does this mean 10 configurations per task or 1000 configurations per task? A search space with 10 configurations in it seems highly unrealistic.~~
- ~~Critical: please provide clarification on the Scikit-CC18 search space.  Namely, what is the size of the algorithm space A?~~
- ~~Critical: I am concerned about the practical application of MASIF to real world search spaces and historical experiments.  To address this concern, I think it is critical to run the additional experiments suggested below:~~
  - ~~Please perform a scaling study of MASIF with larger sets of hyperparameters, e.g. 1000, it seems like experiments only go up to 170 for LCBench.~~
  - ~~Please perform a study of MASIF with incomplete hyperparameter sets for some tasks, e.g. size of A is 1000 but tasks observe learning curves for say 100 configs.~~

**Strengths And Weaknesses:**

Strengths:
- Transfer learning for algorithm selection is an important problem that can benefit from flexible, multi-fidelity approaches.

Weaknesses:
- MASIF requires limiting algorithm configuration space to a fixed set.  This can be a significant limitation for large search spaces where it is difficult to get good coverage with a fixed set.  It also limits the applicability of MASIF to existing learning curves from fixed sets and largely precludes the use of MASIF on learning curves sampled randomly from continuous search spaces.  This is more a limitation of the algorithm selection setting studied than the methodology itself.
- The performance of MASIF is not significantly better than non-metalearned baselines while requiring significant meta-training data to learn the MASIF model.  While MASIF does well on Synthetic and TaskSet, Scikit-CC18 and LCBench are arguably more reflective of the types of tuning tasks a data scientist would perform in practice. On these, benchmarks, there isn't that large of a gap (~0.02) between MASIF and IMFAS and SH.  In terms of meta-training data, the cost I'm referring to is that of collecting learning curves for tasks in the meta-training set which can be expensive if starting from scratch.
- ~~I have many reservations about the experiments.~~  In particular, the baselines should be stronger. ~~SH results look questionable for TaskSet, there are a lot of seemingly arbitrary decisions on the included tasks and hyperparameters for the benchmarks studied.~~
- ~~The writing is confusing and can be improved.   In particular, I highly recommend the authors change the phrasing "myopic/myopia". I also find claims about new multi-fidelity meta-learning setting to not be well supported (see [Multi-task Learning Curve Forecasting Across
Hyperparameter Configurations and Datasets](https://2021.ecmlpkdd.org/wp-content/uploads/2021/07/sub_1011.pdf)).~~

**Post author response**
Thanks for responding to my questions and comments and I recognize I made a few unfair comparisons between the algorithm selection setting and HPO.  I have edited my review accordingly.

---

> ### Author Response · Authors · 2023-01-24
> **Answer to reviewer bHsm based on the first revision.**
>
> Dear bHsm,
>
> Thank you for your review. Unfortunately, we believe that there is a critical misunderstanding of our intended approach. In contrast to algorithm configuration and hyperparameter optimization, the objective of algorithm selection (AS) and thus also of MASIF is to efficiently select the best performing algorithm from a **small set** of target algorithms. We refer to [Schede et al. (JAIR 2022), A Survey of Methods for Automated Algorithm Configuration](https://arxiv.org/pdf/2202.01651.pdf) for a detailed discussion of the differences. This has several implications for the requested changes we would like to discuss point by point.
>
> > 1. MASIF requires limiting algorithm configuration space to a fixed set. This can be a significant limitation for large search spaces where it is difficult to get good coverage with a fixed set.
>
> We agree that the small, fixed set of target algorithms in AS is a limitation compared to HPO. Nevertheless, we believe that our proposed approach is in line with the practical problem of data scientists to efficiently select a promising ML algorithm with little available performance data.
>
> > 2. It also limits the applicability of MASIF to existing learning curves from fixed sets and largely precludes the use of MASIF on learning curves sampled randomly from continuous search spaces.
>
> As said above, yes, MASIF is limited to a fixed set of target algorithms and their learning curves. However, it can interpret any budget allocation allowing learning curves with very different sample points
>
> > 3. The performance of MASIF is not significantly better than non-metalearned baselines while requiring significant meta-training data to learn the MASIF model.
>
> We would like to emphasize that according to Figure 3, MASIF is in fact significantly better on all four benchmarks compared to “Parametric” (i.e., parametrically fitted learning curves) and better than successive halving on three benchmarks while being on par on the remaining one. MASIF is the only approach that showed consistent good and robust performance on all benchmarks, in particular for the fidelity range <0.5 of practical interest to a data scientist. Therefore, we disagree with your assessment.
>
> > 4. I have many reservations about the experiments. In particular, the baselines should be stronger, SH results look questionable for TaskSet, there are a lot of seemingly arbitrary decisions on the included tasks and hyperparameters for the benchmarks studied.
>
> We detailedly discuss this below the requested changes. We emphasize here again that we believe that the reservations about the experiments are mainly due to the misunderstanding concerning the difference between algorithm configuration and algorithm selection.
>
> >5. The writing is confusing and can be improved. In particular, I highly recommend the authors change the phrasing "myopic/myopia".
>
> Thank you for pointing this out. We revised the wording of the paper already based on the feedback from the other reviewers. Therefore, we would like to ask you whether you have read the original submission or the first revision.
> Regarding the use of “myopic/myopia”, we believe that it is in line with the common use of it. For example, the Cambridge dictionary says that [myopia](https://dictionary.cambridge.org/de/worterbuch/englisch/myopia) refers to “a condition in which someone cannot clearly see things that are far away” and is a direct synonym for short-sightedness. We revised the paper to clarify this, see Footnote 2, and highlight that “myopic/myopia” are typical terms also used in the following papers:
> * [Flennerhag et al. (ICLR 2022), Bootstrapped meta-learning](https://arxiv.org/pdf/2109.04504.pdf)
> * [Xu et al. (NeurIPS 2018), Meta-Gradient Reinforcement Learning](https://proceedings.neurips.cc/paper/2018/file/2715518c875999308842e3455eda2fe3-Paper.pdf)
> * [Krueger et al. (CoRR 2020), Hidden Incentives for Auto-induced Distributional Shift](https://proceedings.neurips.cc/paper/2018/file/2715518c875999308842e3455eda2fe3-Paper.pdf)
> * [Dery et al. (NeurIPS 2022), Multi-step Planning for Automated Hyperparameter Optimization with OptFormer ](https://arxiv.org/abs/2210.04971)

---

> > ### Author Response · Authors · 2023-01-24
> > **Continuation (1) : Answer to reviewer bHsm based on the first revision**
> >
> >
> > >6.  I also find claims about new multi-fidelity meta-learning setting to not be well supported (see Multi-task Learning Curve Forecasting Across Hyperparameter Configurations and Datasets).
> >
> > There is indeed some similarity, but also multiple strong distinctions that we detail in Appendix A.4 of the manuscript. These also obstruct us from comparing against their method.
> > 1. First and foremost, Multi-LCNet  addresses an HPO setting, whereas MASIF targets  algorithm selection.
> > 2. Second, MASIF is more broad in scope, allowing us to be agnostic as to what an algorithm is. That is, Multi-LCNet requires auxiliary learning curves, such as gradient information of particular layers which might not be available to all candidates we can consider.
> > 3. Third, in the augmented and novel AS setup we propose, the selector can leverage  the gathered “test-time” evidence on a set of new configurations. This is akin to “peaking” and in this sense Multi-LCNet is similar in setup, but Multi-LCNet does not share the evidence gained when peaking for one configuration with that of another curve. They are considered independent observations, although the peaking may have gained information elucidating likely aspects of that curve.
> >
> > > 7. Critical: add better baselines like BOHB and Learning Curve Prediction with Bayesian Neural Networks.
> >
> > a. BOHB is a Hyperband extension that uses Bayesian Optimization for sampling new configurations. This does not apply in our setup since we only choose which algorithm to run on what fidelity – the sampling process is not covered by MASIF and only mentioned in the future work section. Thus, a comparison against BOHB is impossible.
> >
> > b. Thanks for pointing out a comparison with LCNet (Learning Curve Prediction with Bayesian Neural Networks). We started to look into it and hope to provide a comparison before the decision deadline in two weeks. From a theoretical point of view, MASIF has several advantages: (i) jointly modeling all available learning curves, (ii) taking dataset meta-features into account and (iii) taking algorithm information into consideration.
> >
> > > 8. Critical: run a version of SH/BOHB that uses the same resources as that given to MASIF. For example, MASIF sees learning curves trained on a total of # configurations * fidelity, which will be the resources allocated to SH/BOHB.
> >
> > We believe that the current experiments show ample evidence that MASIF outperforms SH. In fact, MASIF already performs very strong on zero-fidelity, i.e., MASIF does not require any substantial compute. In detail, MASIF on zero fidelity performs better than SH on full fidelity on both Synthetic and Task Set. On LCBench MASIF on zero fidelity performs better than SH on fidelities < 0.6. On Scikit-CC18 both approaches are on par from the beginning.
> >
> > > 9. Critical: please check result for SH on Taskset. SH is able to learn something useful on that benchmark according to Figure 3 of this paper.
> >
> > According to personal communication with the authors of [Dynamic and Efficient Gray-Box Hyperparameter Optimization for Deep Learning (NeurIPS 2022)](https://arxiv.org/pdf/2202.09774.pdf), they removed diverging learning curves from Taskset. This dramatically improves the rank correlation and thus SH performs better in their experiments. We believe that this is not a reasonable assumption in practice where diverging training can happen; therefore, we kept those diverging curves and thus SH performs worse in our paper.
> >
> > > 10. Critical: please evaluate MASIF on Taskset with all datasets included instead of just the selected tasks with MLP on image datasets.
> >
> > We thank you for this suggestion. We believe that there is little meta-knowledge to gain by looking at NLP performance data and transferring it to image datasets. Furthermore, we assume that it is readily available information to which data modality a dataset belongs to (e.g, images or NLP).
> > Nevertheless, we are in the process of running further experiments on the NLP learning curves in Taskset to provide more empirical evidence that MASIF can not only perform well n tabular and image data, but also on NLP tasks.
> >
> > > 11. Critical: please provide clarification on TaskSet search space. The meta-dataset is described as 1000 configurations on 100 tasks in the appendix, does this mean 10 configurations per task or 1000 configurations per task? A search space with 10 configurations in it seems highly unrealistic.
> >
> > In the current setting, we consider 1000 configurations per task, i.e. each of the 100 tasks has the same 1000 configurations. Given that we are currently running other task families as well, we will shortly add their respective specifications.
> >
> > We again note that such a large amount of target algorithms (i.e., configurations here) is not the main use case to MASIF, but is useful for understanding its flexibility and scalability, demonstrating the potential of MASIF.
> >
> > We have additionally added further details about our setup in Appendix A.1.

---

> > > ### Author Response · Authors · 2023-01-24
> > > **Continuation (2) : Answer to reviewer bHsm based on the first revision**
> > >
> > >
> > > > 12. Critical: please provide clarification on the Scikit-CC18 search space. Namely, what is the size of the algorithm space A?
> > >
> > > As stated in Appendix A.1 (Scikit-CC18), it features 16 algorithms. Please refer to Appendix A.1 for more details on the benchmark.
> > >
> > > > 13. Critical: I am concerned about the practical application of MASIF to real world search spaces and historical experiments. To address this concern, I think it is critical to run the additional experiments suggested below:
> > >
> > > Please consider the answer provided at the top before we delve into the discussion of the following two experiments.
> > >
> > > > 13.1 Please perform a scaling study of MASIF with larger sets of hyperparameters, e.g. 1000, it seems like experiments only go up to 170 for LCBench.
> > >
> > > [Tornede et. al. (Discovery Science 2020) Extreme Algorithm Selection with Dyadic Feature Representation](https://arxiv.org/pdf/2001.10741.pdf) argues that large spaces are usually problematic in AS, particularly due to data availability; i.e. the algorithms are not evaluated on all datasets. This is also in line with the practicality of AS and data scientists’ workflows; considering only a set of known to be well performing algorithms on related tasks. In fact, most research on algorithm selection only considers up to tens of algorithms, see e.g.[“ASlib: A Benchmark Library for Algorithm Selection” by Bischl et al. (AIJ 2016)](https://www.sciencedirect.com/science/article/pii/S0004370216300388). We mainly follow this line of arguments, but show in our Taskset experiments that MASIF can also deal with 1000 algorithms, i.e., hyperparameter configurations.
> > >
> > > > 13.2. Please perform a study of MASIF with incomplete hyperparameter sets for some tasks, e.g. size of A is 1000 but tasks observe learning curves for say 100 configs.
> > >
> > > Our interpretation of this request is that you would want to see how MASIF behaves, when only a few algorithms are evaluated to a considerable extent, while the others are allocated zero fidelity.  Although it would in theory be possible to perform an expensive line of experiments on that, there would be no baseline MASIF could be directly compared against. Learning curve predictors and SH are incapable of extrapolating zero fidelity. Only SATzilla seems to be sensible from our point of view, but MASIF outperforms it at zero fidelity already.

---

> ### Author Response · Authors · 2023-02-03
> **Answer to reviewer bHsm on the second revision.**
>
> Dear bHsm,
>
> We would like to give you an update on the current state of your requests:
>
> >Strengthen: please evaluate MASIF on Taskset with all datasets included instead of just the selected tasks with MLP on image datasets.
>
> We have been looking into it and already have some preliminary results for the CNN subset of Task Set that contains the learning curves of 1000 algorithms on 76 datasets. Unfortunately, the parametric baseline has timed out because it is an extensive set of learning curves that need to be fitted by all of the candidate shapes on all of the fidelity levels independently. After restarting it, we currently have roughly half of its seed-fold combinations, which you can find [here](https://anonymous.4open.science/r/MASIF-824D/img/TaskSet_partial_CNN.png). Overall the results are consistent with those previously observed in Figure 3 with the MLP subset, showing that MASIF outperforms all other baselines.
>
> > Strengthen: add better baselines like BOHB and Learning Curve Prediction with Bayesian Neural Networks. I understand that the problem solved by BOHB is different since it can sample from the search space but this is a strictly harder problem than selecting from a finite set as MASIF is doing. I can imagine running BOHB with different fidelities and giving it the same number of tries as the finite set considered by MASIF and report the final performance of the best found configuration.
>
> We are currently running the LCNet experiments and will post the results hopefully by mid of the next week. We have adapted the training and testing protocol of LCNet to meet the algorithm selection setup in the following ways:
> 1. Each dataset is fed to LCNet independently and only up to the current fidelity horizon of the slice evaluation protocol.
> 2. Since LCNet is intended for HPO, it requires the hyperparameters of the learning curves. Therefore, we naturally can only compare against it on LCBench.
>
> Unfortunately, your suggestion regarding BOHB does not provide a fair comparison. Our primary goal is to interpret provided fidelity information efficiently based on the available meta-knowledge. The chosen baselines seek to illustrate this gain in efficiency.
> Your suggestion on the other hand implies that we compare a new hyperparameter configuration against the existing set of candidate algorithms, which is not a fair comparison to make and is not covered by the algorithm selection scenario. Whether the one or the other can be considered as the harder problem depends on the set of candidate algorithms vs the landscape of the HPO problem.

---

> > ### Author Response · Authors · 2023-02-21
> > **Latest Revision and Additional Results**
> >
> > Dear bHsm,
> >
> >
> > >Strengthen: please evaluate MASIF on Taskset with all datasets included instead of just the selected tasks with MLP on image datasets.
> >
> > We are glad to finally report to you the results of your requested experiments. We have included the
> > In particular, you will find the results
> >
> > 1. of the CNN-NLP Task Set subset including the complete set of parametric learning curves can be found in the appendix. For the details of the NLP and image subsets, please refer to the added paragraph in Appendix A.1. We are happy to report that the results obtained from the CNN-NLP Task Set subset are consistent with the results we obtained on the MLP-Image subset we previously evaluated on, with the exception that the parametric learning curves do not manage to close the performance gap given the final fidelity.
> >
> > 2. of the adapted LCNet experiments can be found in the paper’s result section, shown in the LCBench plot.
> > We would like to highlight that LCNet computes its predictions solely on the hyperparameter configuration of the algorithm and the associated fidelity. This information is only available in the LCBench benchmark since it originates from a hyperparameter optimization problem that we subsampled to a known and discrete set of hyperparameters to meet the algorithm selection setup. Despite this limitation, the results on LCBench indicate that the probabilistically weighted ensemble of parametric learning curves works well on this relatively simple benchmark. Within the interval of roughly [0.2, 0.4] - that is considerably later than the meta-aware algorithms - it achieves a competitive performance. However, like all meta-agnostic algorithms on this benchmark, it initially requires some adaptation to the incoming fidelity information. Interestingly, its estimated ranking deteriorates slightly beyond the 0.4 thresholds.
> > Overall, we conclude that MASIF is a competitive choice on this benchmark due to its meta-awareness, allowing it to predict an almost ideal ranking on this benchmark from the start at no additional cost.

---

### Review · Reviewer_7vrQ · 2023-01-29

**Summary Of Contributions:**

This paper proposes an approach based on a meta-learned Transformer to jointly use multi-fidelity and task feature information to perform algorithm selection. It uses a Transformer trained on part of a meta-dataset of algorithmic selection tasks to train a ranking predictor, which it then evaluates in combination with successive halving on a variety of tasks.

**Audience:**

Yes

**Claims And Evidence:**

Yes

**Requested Changes:**

Addressing the line comments below would strengthen the work, but are largely easy/not critical. I do believe the last point (line 346) on lack of comparison with the latest SH methods is the biggest in terms of potential greater impact of the paper. I also think it is important to at least mention the OptFormer work and compare methodology (I understand a direct empirical comparison might be difficult).

34: perhaps the survey paper of Mohr & van Rijn (2022) should be linked here to summarize the field, not a single unpublished paper by the same authors?
43: “with few exceptions, such as (van Rijn 44 et al., 2015; Klein et al., 2017; Baker et al., 2018; Long et al., 2020)” - it seems weird to say “few exceptions” here when only two works (and by the same exact authors, Mohr & van Rijn) were pointed to as examples of the general approach.
100: are these rankings strict or partial orderings?
106: Outside a multi-objective setting, why do we care about the ranking error, e.g. why does it matter if we rank the second-worst and worst algorithms incorrectly?
186: unclear if a lot of discussion of positional encoding is necessary; while the Transformer model indeed ignores positional information, the solution you use is the one used in the original paper to solve the same exact problem.
343: what is the difference between the present paper and IMFAS, other than going from LSTM to Transformers? Could it not also be combined with SH?
346: a variety of strong, widely used approaches have been built on top of SH, including Hyperband, ASHA, and BOHB. Comparison with those would strengthen the paper


**Strengths And Weaknesses:**

Strengths:
- Combining both multi-fidelity information and learning curve prediction is an interesting challenge and does not seem to have been fully and satisfactorily handled in the past.
- The paper presents a fairly natural solution to the problem, and the presentation is for the most part well-done, if sometimes a bit tedious.
- The solution is evaluated on a range of meta-datasets.

Weaknesses:
- There are numerous seemingly related methods that could have been compared to or at least discussed. On the comparison side, successive halving is not really the leading method in multi-fidelity optimization, with more advanced variants like BOHB and ASHA being widely used. On the discussion (and perhaps comparison) side, what is the relationship between this paper and an approach like OptFormer (Chen et al., NeurIPS 2022)?
- The proposed method only seems obviously better at low fidelity, and the improvement due to SH is inconsistent (e.g. it is worse on TaskSet), so there is no single solution that emerges.

---

> ### Author Response · Authors · 2023-02-03
> **Second Revision.**
>
> Dear 7vrQ,
>
> Thank you very much for your detailed review. We have a few reservations regarding the requested experiments we would like to discuss jointly for the three comments below
>
> >There are numerous seemingly related methods that could have been compared to or at least discussed. On the comparison side, successive halving is not really the leading method in multi-fidelity optimization, with more advanced variants like BOHB and ASHA being widely used. On the discussion (and perhaps comparison) side, what is the relationship between this paper and an approach like OptFormer (Chen et al., NeurIPS 2022)?
>
> >346: a variety of strong, widely used approaches have been built on top of SH, including Hyperband, ASHA, and BOHB. Comparison with those would strengthen the paper
>
> >Addressing the line comments below would strengthen the work, but are largely easy/not critical. I do believe the last point (line 346) on lack of comparison with the latest SH methods is the biggest in terms of potential greater impact of the paper. I also think it is important to at least mention the OptFormer work and compare methodology (I understand a direct empirical comparison might be difficult).
>
> Thank you for bringing this up. Since this seems to cause misunderstandings, we have added a few dedicated paragraphs in the related work section regarding fidelity-aware Hyperparameter Optimization(-al) (HPO) methods.
>
> Crucially, we hope to clarify to the reader that we are performing algorithm selection and not HPO. In algorithm selection, we seek to efficiently select the best-performing algorithm from a **small set** of (known) target algorithms. We refer to [Schede et al. (JAIR 2022), A Survey of Methods for Automated Algorithm Configuration](https://arxiv.org/pdf/2202.01651.pdf) for a detailed discussion. This implies,
> - that we are not able to select new configurations via e.g. BO or random search in the case of HB. So neither *HB* nor *BOHB* is an applicable competitor in our setting.
> - *ASHA* as an asynchronous extension to SH does not provide a reasonable benefit here as it merely allows it to exploit parallel resources more aggressively and base its decisions on even more premature information than SH. ASHA serves as just yet another example of a budget allocation strategy that can benefit from MASIF’sinterpretation model as outlined in Lines 292 ff.
> - *OptFormer*. This model is aimed at active sequential decision-making akin to BO in the domain of HPO, but combats a different kind of myopia. While we are combating myopia resulting from premature evaluation induced by multi-fidelity, they are concerned with the short-sightedness resulting from the sequential decision process across hyperparameter configurations. Both models address distinct tasks.
>
> In summary, we believe that there is no sensible way to compare against the baselines you suggested while staying in the setting MASIF is designed for.
>
>
> >The proposed method only seems obviously better at low fidelity, and the improvement due to SH is inconsistent (e.g. it is worse on TaskSet), so there is no single solution that emerges.
>
> We agree that MASIF is already better at low fidelity. This is where most computation can be saved and details the great benefit of meta-learning (jointly) as performed by MASIF. However, we are unsure what exactly you are referring to by the “improvement due to SH”. We assume that you are referring to the MASIF+SH variant. In a way, the MASIF+SH variant is more limiting than MASIF on its own, as it prescribes a specific budget allocation strategy and continuously reduces the set of algorithms to choose from. Thus, it can happen that MASIF+SH performs worse than MASIF.
>
>
> >34: perhaps the survey paper of Mohr & van Rijn (2022) should be linked here to summarize the field, not a single unpublished paper by the same authors?
>
> Thanks for catching this citation error. Indeed, we already cited the paper at a different place and have corrected this error.
>
>
> >43: “with few exceptions, such as (van Rijn 44 et al., 2015; Klein et al., 2017; Baker et al., 2018; Long et al., 2020)” - it seems weird to say “few exceptions” here when only two works (and by the same exact authors, Mohr & van Rijn) were pointed to as examples of the general approach.
>
> We use the comprehensive survey by [Mohr & van Rijn (Corr 2022), Learning Curves for Decision Making in Supervised Machine Learning — A Survey ](https://arxiv.org/pdf/2201.12150.pdf) as a source to which we can point for all the papers related to the usage learning curves. This includes the approaches that we characterize as myopic in section 4.2 Parametric Learning Curve Predictors. Thus, the list of exceptions is in fact rather short compared to all papers reviewed in the survey.

---

> > ### Author Response · Authors · 2023-02-03
> > **Second Revision. Continuation**
> >
> >
> > >100: are these rankings strict or partial orderings?
> >
> > For abstraction across various models, we deliberately do not say this in the preliminaries. Both are possible depending on the selection use-case and merely constrain the loss/evaluation function. In our experiments, MASIF learns a strict ranking overall algorithms.
> >
> > >106: Outside a multi-objective setting, why do we care about the ranking error, e.g. why does it matter if we rank the second-worst and worst algorithms incorrectly?
> >
> > If we only cared about a single snapshot of a budget allocation, then the top-1 regret is indeed the only metric that matters to a user and one could resort to a classification loss instead. However, as our MASIF+SH variant already illustrates; one of the most immediate extensions in future work is using it for scheduling (i.e. multiple subsequent budget allocation snapshots) in an elaborate manner. Here, for instance, MASIF+SH decides on the termination of an algorithm based on the lower half of the current ranking. Notably, the gathered evidence on the new dataset is never forgotten but accumulated.
> >
> > >186: unclear if a lot of discussion of positional encoding is necessary; while the Transformer model indeed ignores positional information, the solution you use is the one used in the original paper to solve the same exact problem.
> >
> > We deem the three sentences in Lines 186-189 describing the initial positional encoding referencing Vaswani rather short. We have been slightly more verbose in the latest revision regarding the second positional encoding in Line 239, which serves a different purpose and has been clarified now.
> >
> > >343: what is the difference between the present paper and IMFAS, other than going from LSTM to Transformers? Could it not also be combined with SH?
> >
> > There are three technical parts to this answer:
> > 1. As you said; ours has a transformer as its prediction module and thus does not fall victim to well-known autoregressive issues.
> > 2. The masking scheme, native to transformers, is capable of interpreting any budget allocation not limited to the slice evaluation protocol, as the MASIF+SH variant demonstrates. IMFAS is tied to this protocol. Therefore, in its current form, it is not applicable for a combination with SH.
> > 3. The dataset meta-features are an optional drop-in feature in MASIF. IMFAS, on the other hand, requires it as initialisation to its LSTM’s hidden state. If a dataset lacks meta-features, there is no straightforward way of adapting it, hence limiting its applicability.

---

### Author Response · Authors · 2023-01-13
**General response & first revision.**

We would like to thank all reviewers for their detailed and helpful reviews. We strongly believe that your comments helped us to improve the paper. Before answering each reviewer’s comments in detail, we would like to detail some general changes below.

Generally, we took reviewer **UzPx**’s remark regarding verbosity to heart and edited the formulations of our text to make them more concise and more easily readable. These changes do not change the overall structure of the text but they are supposed to clear up the formulation for ease of reading. We also added ablations regarding the architectural design for algorithm-/dataset meta-features in Appendix A.4.1 and regarding hyperparameters of particular interest in Section 3.4.

We would like to note a small documentation error on our part in the appendix we corrected in this revision: LCBench was not subsampled by a Latin Hypercube, but instead by the union of the top-10 performing algorithms per dataset. This is the reason for the 170 algorithms considered. The validity of the results do not change because of that.

As reviewer **p2g7** suggested, we

1. moved LCBench from the appendix into the results section, including its discussion as to why it is not an applicable benchmark for future studies.
2. added the hyperparameters of our architecture into the appendix, to allow for the re-implementation of our architecture.

We uploaded a new revision of our paper and marked all changes in blue s.t. the reviewers can easily track all the changes. (We note that the action editor allowed us to start with the rebuttal and a new revision although the third review is still pending.)

---

### Decision · Action_Editors · 2023-03-07

**Recommendation:** Accept as is

**Comment:**

This paper proposes an approach to AS that uses transformers meta-learned on task features, algorithm features, and learning curves at different fidelities in order to learn to rank algorithms for a given task. This use of transformers for algorithm selection is quite original, and removing the need for parametric assumptions on the learning curve is very interesting. The reviewers had a number of concerns around verbosity, ablations, and additional benchmarks that were largely resolved through the discussion process.

There was some reservation during the final discussion about the usefulness of the approach in terms of the AS setup. The argument is that the use of a fixed set of hyperparameters is not very representative of the workflow of a typical data scientist. A comparison or extension to multi-fidelity HPO is certainly worth thinking about, however it was felt that the paper is still worth publishing with this setup as it may pave the way for future work in this area.

**Audience:**

Yes, this is of interest to the AS community and possibly opens up new approaches for HPO as well.

**Claims And Evidence:**

Yes, the claims in the paper are generally backed up by experiments across four different benchmarks and using a fair number of baseline approaches for AS. Ablations for the method are given in the appendix.